# Interval-Valued Fuzzy Multi-Criteria Decision-Making with Dependent Evaluation Criteria for Evaluating Service Performance of International Container Ports

**Yu-Jie Wang [1], Li-Jen Liu [2] and Tzeu-Chen Han [1],***

[1] Department of Shipping and Transportation Management, National Penghu University of Science and Technology, Penghu 880, Taiwan; knight@gms.npu.edu.tw
[2] Dinos International Corporation, Taipei 104, Taiwan; jamesliu@dinosintl.com.tw
* Correspondence: tchan@gms.npu.edu.tw; Tel.: +886-69264115 (ext. 5515)

**Abstract:** Due to COVID-19 barriers, the needs of international container ports have become more important than in the past. Therefore, it is very critical and essential for the scientific developments of port-logistics. To gain the scientific developments of port logistics, effective and efficient evaluation methods for decision-making are indispensable, especially for assessing service performance of international container ports based on dependent evaluation criteria (DEC). Among numerous decision-making methods, technique for order preference by similarity to ideal solution (TOPSIS) was often expanded under fuzzy environments into fuzzy multi-criteria decision-making (FMCDM) to preserve imprecise messages. The FMCDM was able to be associated with quality function deployment (QFD) into a hybrid method to solve problems with DEC. To gain more messages, QFD and TOPSIS are combined and then expanded under interval-valued fuzzy environment (IVFE) to solve a FMCDM problem with DEC. Practically, evaluating service performance of international container ports in Taiwan and the surrounding sea areas is considered a problem with DEC because the related evaluation criteria are partially connected. By the hybrid method of combining QFD with TOPSIS under IVFE, international container ports with DEC are effectively and efficiently evaluated for service performance, and more insights are gained than the past for establishing essential fundamentals in recent scientific developments of port logistics on account of breaking down COVID-19 barriers.

**Keywords:** dependent evaluation criteria (DEC); international container ports; interval-valued fuzzy numbers (IVFNs); quality function deployment (QFD); technique for order preference by similarity to ideal solution (TOPSIS)

## 1. Introduction

Since the outbreak of COVID-19, the tasks of international container ports for transportation have become more essential for world economics. Before COVID-19, passenger transportation and cargo freight might have been equally important; however, recently, people are more often confined in certain regions and, thus, are more reliant on purchasing goods to satisfy life requirements than previously. It is evident that, globally, numerous cargos in the entrepot trade are needed and have to be transported via international container ports. It is said that recent scientific developments are very important for port logistics [1] because of equipment and vessel shortages, total capacity decline, port congestion, and cost increase during COVID-19 outbreaks. In Taiwan and surrounding sea areas, there are some important container ports including Hong Kong, Shanghai, Kaohsiung, Shenzhen, Singapore, Tokyo, Pusan, Klang, Manila, Laem Chabang, Oingdao, and Tanjung Priok [2]. The efficiency measurement of these international container ports will be essential for developing port logistics in Asia and the Pacific regions. Practically, 13 evaluation criteria are taken into consideration: tugboat operation, untwisting rope operation, pilot operation,

stevedoring efficiency, low damage rate for goods, waiting time for unloading, working and service flexibility, application service process, service personnel ability, service personnel attitude, advisory services, harbor rates, and stevedoring rates. Due to the close connection between partial evaluation criteria, an effective and efficient evaluation method is indispensable for evaluating service performance of international container ports based on dependent evaluation criteria (DEC). To DEC, some past approaches evaluated the related problems with analytic network process (ANP) [3] or DEC transformed into independent evaluation criteria (IDEC). However, ANP was limited in data specifications and complicated questionnaires, whereas DEC transformed into IDEC might add some computations, such as factor analysis [4] and criteria weights reassessed. In this paper, we desire to use a hybrid method [5], merging several techniques to evaluate decision-making problems with DEC, and avoid the drawbacks mentioned above. We discuss the development of the hybrid method as follows:

Evaluation should be one of the important decision-making issues, and some criteria considered belong to multi-criteria decision-making (MCDM) [6]. A MCDM problem is generally shown as:

$$
G = \begin{array}{c} \\ A_1 \\ A_2 \\ \vdots \\ A_m \end{array}
\begin{array}{c} C_1 \; C_2 \; \ldots \; C_n \\
\begin{bmatrix}
G_{11} & G_{12} & \cdots & G_{1n} \\
G_{21} & G_{22} & \cdots & G_{2n} \\
\vdots & \vdots & \cdots & \vdots \\
G_{m1} & G_{m2} & \cdots & G_{mn}
\end{bmatrix}
\end{array}
\tag{1}
$$

and

$$
W = [W_1, W_2, \ldots, W_n]
\tag{2}
$$

where $A_i$ denotes the $i$th alternative, $C_j$ indicates the $j$th criterion, $G_{ij}$ is the rating of $A_i$ on $C_j$, and $W_j$ is the weight of $C_j$ for $i = 1, 2, \ldots, m; j = 1, 2, \ldots, n$. In the previous MCDM problems [7], some with imprecise messages on evaluation are regarded as fuzzy MCDM (FMCDM) problems [8] because these messages are commonly assessed by linguistic variables [9,10] and then displayed by fuzzy numbers [11]. Practically, imprecise messages, including alternative ratings and criteria weights, are indicated by linguistic variables, and then represented by fuzzy numbers in FMCDM problems.

Moreover, FMCDM problems [12] having IDEC were widely described. However, some with DEC might be rarely mentioned due to complex computation. In recent years, FMCDM problems with DEC were gradually discussed. To overcome complex computations of DEC, quality function deployment (QFD) [13] is utilized in aggregating customer requirements and technical solutions to derive criteria weights. In QFD [14], customer requirements indicate user opinions, and technical solutions denote professional viewpoints. In QFD, the customer requirements are expressed in an important level matrix, and the relationships between customer requirements and technical solutions are presented through a relation matrix. According to the two matrices above, the importance and relationship of DEC are rationally displayed, and then criteria weights are derived by QFD. In addition, QFD can be extended under fuzzy environment into fuzzy QFD (FQFD) [15]. In FQFD, entries of the two matrices are displayed by fuzzy numbers. Owing to data characteristics, the computation of FQFD [16,17] is complicated for combining the two matrices. Liang's approach [18] is illustrated to describe the complicated computation.

In Liang's FQFD [18], the important level matrix and the relation matrix, through matrix multiplication, were combined into a criterion weight. Further, two triangular or trapezoidal fuzzy numbers were integrated into a weighted relationship rating by multiplying the two matrices above, and the weighted relationship rating based on the fuzzy extension principle [11] was a pooled fuzzy number (PFN). All weighted relationship ratings within each technical solution are summarized and then averaged to form a criterion weight that is also a PFN. Practically, the fuzzy calculation [19] of the yield of the weighted values is difficult owing to the multiplication of trapezoidal fuzzy numbers. In fact, these previous computations used in interval-valued fuzzy numbers (IVFNs) [20,21] were harder

than other fuzzy numbers. The aggregation computation of pooled fuzzy numbers (PFNs) in the extension of QFD (i.e., FQFD) to form criteria weights was critical, especially for FMCDM with DEC, regardless of whether ratings or weights of alternatives were IVFNs or not.

In recent years, some proposed FMCDM having IDEC under an interval-valued fuzzy environment (IVFE) [22,23] in order to obtain more information than other versions, such as triangular or trapezoidal fuzzy numbers. However, FMCDM with DEC for IVFNs was rarely discussed due to difficult computations of DEC. Herein, QFD and technique for order preference by similarity to ideal solution (TOPSIS) are expanded under IVFE [24] to resolve the multiplication tie of fuzzy numbers for obtaining more messages in FMCDM with DEC. TOPSIS [6] is a famous method and often expanded under fuzzy environment into FMCDM [25,26]. The underlying logic of TOPSIS mainly defines anti-ideal solution and ideal solution in decision-making. The anti-ideal solution maximizes cost criteria and minimizes benefit criteria, whereas the ideal solution maximizes benefit criteria and minimizes cost criteria. Therefore, the optimal alternative within all candidate alternatives has the farthest distance to the anti-ideal solution and the shortest distance to the ideal solution. Moreover, the distances of candidate alternatives to the anti-ideal solution and the ideal solution are aggregated into relative closeness coefficients in TOPSIS. Then, all candidate alternatives are ranked based on their corresponding coefficients.

Some past approaches [8,12,19,27] were useful to TOPSIS [28] extended under a fuzzy environment, but they were utilized in triangular or trapezoidal fuzzy numbers. In addition, Wang [29], using TOPSIS and relative preference relation (RPR), processed related problems under IVFE because these processes were gradually complicated and, thus, more messages needed to be obtained than in the previous approaches. In Wang's [29] approach, decision-making problems through RPR were constructed on IVFNs [30] and solved by FMCDM with IDEC. The RPR [29] between IVFNs is revised from Lee's [31,32] fuzzy preference relation (FPR) between triangular fuzzy numbers. Therefore, TOPSIS based on the RPR is extended under IVFE into interval-valued FMCDM (IVFMCDM) [20,33] with IDEC in Wang's method [29]. QFD and TOPSIS in this paper are extended for IVFMCDM with DEC by a preference relation similar to Wang's [29] RPR. In fact, Wang [34], based on an extended FPR (EFPR) improved from Lee's method [31,32], had associated QFD with simple additive weighting (SAW) [35] for IVFMCDM with DEC to obtain more data. Similar to TOPSIS, SAW is another famous MCDM method, but TOPSIS has the strength in alternative ranking due to the relative closeness coefficients. The relative closeness coefficients for alternatives are in interval [0,1] and, thus, the ranking of alternatives is easy. Nevertheless, QFD extended for IVFNs in Wang's approach [34] is still an important reference for this paper. Based on the above, QFD is associated with TOPSIS as a hybrid method for IVFNs to solve IVFMCDM problems with DEC. Regarding scientific developments of port logistics, service performance evaluation of international container ports can be regarded as an IVFMCDM problem with DEC. Therefore, it is suitable for IVFNs to be applied as the hybrid method for evaluation.

To be clear, related rationales of IVFNs are expressed in Section 2. In Section 3, the extensions of QFD and TOPSIS for IVFNs in decision-making are presented to solve IVFM-CDM problems with DEC. A numerical example about service performance evaluation of international container ports with DEC is calculated in Section 4. Eventually, conclusions are described in the final section.

## 2. Mathematical Rationales

In this section, related rationales of IVFNs [11] are described as follows.

**Definition 1.** *According to the concept of interval-valued fuzzy sets (IVFSs) [30], an interval-valued fuzzy set(IVFS) A on $(-\infty, \infty)$ is defined as:*

$$A = \{x, [\mu_{A^L}(x), \mu_{A^U}(x)]\}, x \in (-\infty, \infty), \mu_{A^L}, \mu_{A^U} : (-\infty, \infty) \to [0, 1], \quad (3)$$

$$\mu_{A^L}(x) \leq \mu_{A^U}(x), \forall x \in (-\infty, \infty), \mu_A(x) = [\mu_{A^L}(x), \mu_{A^U}(x)], x \in (-\infty, \infty),$$

where $\mu_{A^L}(x)$ is the lower limitation of membership degree and $\mu_{A^U}(x)$ is the upper limitation of membership degree.

Hence, the membership degree of an IVFS $A$ in $x^*$ is expressed by $[\mu_{A^L}(x^*), \mu_{A^U}(x^*)]$ (see Figure 1), where $\mu_{A^L}(x^*)$ and $\mu_{A^U}(x^*)$ denote the minimum and maximum grades of membership in $x^*$.

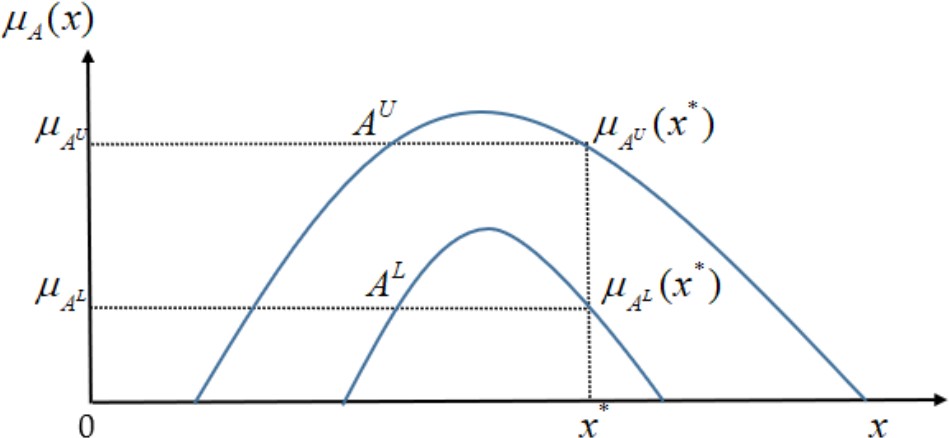

**Figure 1.** An IVFS $A$.

**Definition 2.** *A triangular interval-valued fuzzy number(TIVFN) $A$ (see Figure 2) [36] is denoted as:*

$$A = [A^L, A^U] = [(a_l^L, a_h^L, a_u^L; w_A^L), (a_l^U, a_h^U, a_u^U; w_A^U = 1)], \tag{4}$$

*where $A^L$ and $A^U$, respectively, denote the lower part and upper part of $A$, and $A^L \subseteq A^U$. Moreover, $\mu_A(x)$ is a membership function that indicates the membership grade of $x$, where $\mu_{A^L}(x)$ and $\mu_{A^U}(x)$ are, respectively lower part and upper part of $\mu_A(x)$.*

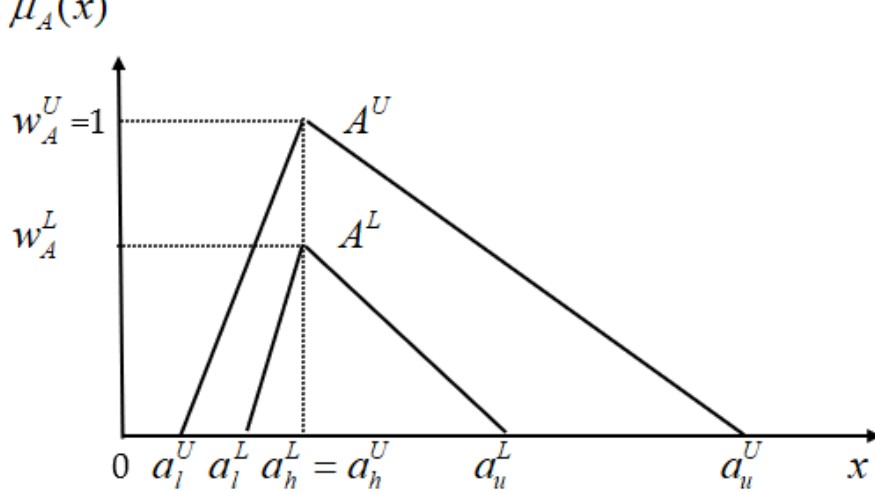

**Figure 2.** A TIVFN $A$.

According to Figure 2, the related lemmas [33] for an interval-valued fuzzy number(IVFN) $A$ are denoted below.

**Lemma 1.** *An IVFN $A$ is a crisp value if $a_l^L = a_l^U = a_h^L = a_h^U = a_u^L = a_u^U$.*

**Lemma 2.** *An IVFN A is a triangular fuzzy number if $A^L = A^U$ (i.e., $a_l^L = a_l^U = a_l$, $a_h^L = a_h^U = a_h$, and $a_u^L = a_u^U = a_u$). A is indicated by a triplet $(a_l, a_h, a_u)$.*

**Lemma 3.** *An IVFN A is a general TIVFN (see Figure 3) presented by $A = [A^L, A^U] = ((a_l^U, a_l^L), (a_h^L = a_h^U), (a_u^L, a_u^U))$ if $w_A^L = w_A^U = 1$ and $a_h^L = a_h^U$.*

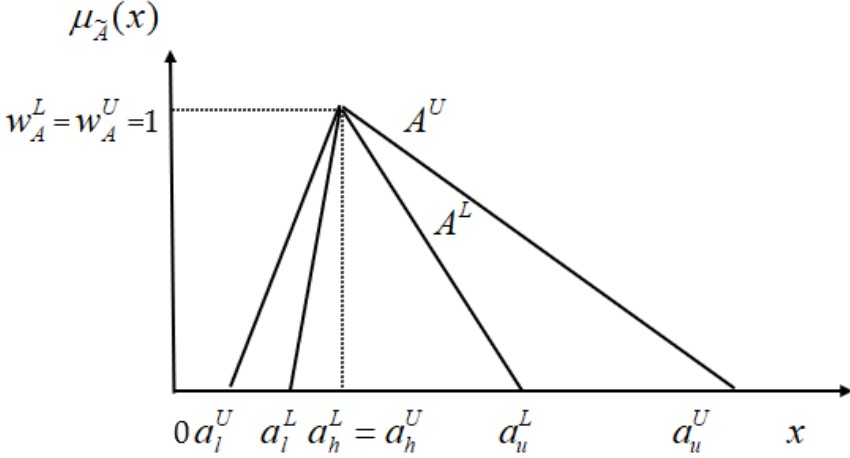

**Figure 3.** A general TIVFN *A*.

Moreover, a general TIVFN *A* with $\mu_{A^L}(x)$ and $\mu_{A^U}(x)$ is defined by:

$$\mu_{A^L}(x) = \begin{cases} \dfrac{x - a_l^L}{a_h^L - a_l^L} & a_l^L \le x \le a_h^L \\[2mm] \dfrac{a_u^L - x}{a_u^L - a_h^L} & a_h^L \le x \le a_u^L \\[2mm] 0 & otherwise \end{cases} \tag{5}$$

and

$$\mu_{A^U}(x) = \begin{cases} \dfrac{x - a_l^U}{a_h^U - a_l^U} & a_l^U \le x \le a_h^U \\[2mm] \dfrac{a_u^U - x}{a_u^U - a_h^U} & a_h^U \le x \le a_u^U \\[2mm] 0 & otherwise \end{cases} \tag{6}$$

where $a_h^L = a_h^U$.

Let $((a_l^U, a_l^L), (a_h^L = a_h^U = a_h), (a_u^L, a_u^U))$ be the general TIVFN *A*. Herein, general triangular interval-valued fuzzy numbers (TIVFNs) are utilized to represent IVFNs below.

**Definition 3.** *Let $\circ$ be an operation on real numbers, such as $+, -, *, \wedge, \vee$, etc. Let $A = [A^L, A^U]$ and $B = [B^L, B^U]$ be two IVFNs. An extended operation $\circ$ generalized from Lee's [31,32] on IVFNs is indicated as:*

$$\mu_{A^L \circ B^L}(z) = \sup_{x,y:z=x \circ y} \{\mu_{A^L}(x) \wedge \mu_{B^L}(y)\} \text{ and}$$
$$\mu_{A^U \circ B^U}(z) = \sup_{x,y:z=x \circ y} \{\mu_{A^U}(x) \wedge \mu_{B^U}(y)\} \tag{7}$$

**Definition 4.** *Let $A = [A^L, A^U]$ be an IVFN. Then, $(A^L)_\alpha^-$, $(A^L)_\alpha^+$, $(A^U)_\alpha^-$, and $(A^U)_\alpha^+$ improved from Lee's methods [31,32] are, respectively, displayed as:*

$$(A^L)_\alpha^- = \inf_{\mu_{A^L}(z) \ge \alpha}(z), \quad (A^L)_\alpha^+ = \sup_{\mu_{A^L}(z) \ge \alpha}(z), \quad (A^U)_\alpha^- = \inf_{\mu_{A^U}(z) \ge \alpha}(z), \text{ and } (A^U)_\alpha^+ = \sup_{\mu_{A^U}(z) \ge \alpha}(z) \tag{8}$$

**Definition 5.** *A FPR R [37,38] is a fuzzy subset of $\Re \times \Re$ with a membership function $\mu_R(A, B)$ representing the preference degree of fuzzy numbers A over B.*

(i)     *R is reciprocal if and only if $\mu_R(A, B) = 1 - \mu_R(B, A)$ for all fuzzy numbers A and B;*
(ii)    *R is transitive if and only if $\mu_R(A, B) \geq \frac{1}{2}$ and $\mu_R(B, C) \geq \frac{1}{2} \Rightarrow \mu_R(A, C) \geq \frac{1}{2}$ for all fuzzy numbers A, B, and C;*
(iii)   *R is a total ordering relation [39,40] if R satisfies both reciprocal and transitive on fuzzy numbers.*

According to *R* [29], *A* is smaller than *B* if $\mu_R(A, B) < \frac{1}{2}$, *A* is larger than *B* if $\mu_R(A, B) > \frac{1}{2}$, or *A* and *B* are no different if $\mu_R(A, B) = \frac{1}{2}$.

**Definition 6.** *An EFPR R' is an extended fuzzy subset of $R \times R$ with membership function $-\infty \leq \mu_{R'}(A, B) \leq \infty$ representing the preference degree of fuzzy numbers A over B [31,32].*

(i)     *R' is reciprocal if and only if $\mu_{R'}(A, B) = -\mu_{R'}(B, A)$ for all fuzzy numbers A and B;*
(ii)    *R' is transitive if and only if $\mu_{R'}(A, B) \geq 0$ and $\mu_{R'}(B, C) \geq 0 \Rightarrow \mu_{R'}(A, C) \geq 0$ for all fuzzy numbers A,B , and C;*
(iii)   *R' is additive if and only if $\mu_{R'}(A, C) = \mu_{R'}(A, B) + \mu_{R'}(B, C)$;*
(iv)    *R' is a total ordering relation if R' satisfies reciprocal, transitive, and additive.*

Based on the EFPR, *A* is smaller than *B* if $\mu_{R'}(A, B) < 0$, *A* is larger than *B* if $\mu_{R'}(A, B) > 0$, or *A* and *B* are no different if $\mu_{R'}(A, B) = 0$.

**Definition 7.** *Let A and B be two general fuzzy numbers. Based on Lee's method [31,32], the EFPR $\mu_{R'}(A, B)$ of A over B is defined as:*

$$\int_0^1 ((A - B)_\alpha^- + (A - B)_\alpha^+) d\alpha \tag{9}$$

**Definition 8.** *For two IVFNs A and B, the EFPR [34] $\mu_{P*}(A, B)$ of A over B is:*

$$\int_0^1 ((A^U - B^U)_\alpha^- + (A^L - B^L)_\alpha^- + (A^L - B^L)_\alpha^+ + (A^U - B^U)_\alpha^+) d\alpha \tag{10}$$

**Lemma 4.** *As $A = ((a_l^U, a_l^L), a_h, (a_u^L, a_u^U))$ and $B = ((b_l^U, b_l^L), b_h, (b_u^L, b_u^U))$ are two TIVFNs, the EFPR $\mu_{P*}(A, B)$ according to Equation (10) is yielded as:*

$$\frac{(a_l^U - b_u^U) + (a_l^L - b_u^L) + 4(a_h - b_h) + (a_u^L - b_l^L) + (a_u^U - b_l^U)}{2} \tag{11}$$

**Definition 9.** *Let $A = ((a_l^U, a_l^L), a_h, (a_u^L, a_u^U))$ and $B = ((b_l^U, b_l^L), b_h, (b_u^L, b_u^U))$ be two TIVFNs. The addition $\oplus$ [17] for A and B is define as:*

$$\begin{aligned} &A \oplus B \\ &= ((a_l^U, a_l^L), a_h, (a_u^L, a_u^U)) \oplus ((b_l^U, b_l^L), b_h, (b_u^L, b_u^U)) \\ &= ((a_l^U + b_l^U, a_l^L + b_l^L), a_h + b_h, (a_u^L + b_u^L, a_u^U + b_u^U)) \end{aligned} \tag{12}$$

**Definition 10.** *The multiplication $\otimes$ [17] of a real number $\beta$ ($\geq 0$) and a TIVFN $A = ((a_l^U, a_l^L), a_h, (a_u^L, a_u^U))$ is defined as:*

$$\beta \otimes A = \beta \otimes \left( (a_l^U, a_l^L), a_h, (a_u^L, a_u^U) \right) = \left( (\beta a_l^U, \beta a_l^L), \beta a_h, (\beta a_r^L, \beta a_r^U) \right) \tag{13}$$

Based on above, the EFPR for a set of IVFNs is expressed as follows:

**Definition 11.** *Let* $S = \{X_1, X_2, \ldots, X_n\}$, *where* $X_j = [X_j^L, X_j^U]$ *is an IVFN for* $j = 1, 2, \ldots, n$. $X' = [X'^L, X'^U]$ *is the reference basis for these IVFNs in EFPR. Then,* $\mu_{P*}(X_j, X')$ *is derived by* $P*$ *as an EFPR to represent the preference degree of* $X_j$ *over* $X'$ *for* $X_1, X_2, \ldots, X_n$. *According to Equation (10),*

$$\mu_{P*}(X_j, X') = \int_0^1 ((X_j^U - X'^U)_\alpha^- + (X_j^L - X'^L)_\alpha^- + (X_j^L - X'^L)_\alpha^+ + (X_j^U - X'^U)_\alpha^+)d\alpha,$$

(14)

*where* $j = 1, 2, \ldots, n$.

**Lemma 5.** *Let* $X' = \left((X'^U_l, X'^L_l), X'_h, (X'^L_u, X'^U_u)\right)$ *be the reference basis of* $X_1, X_2, \ldots, X_n$, *where* $X_j = \left((x^U_{jl}, x^L_{jl}), x_{jh}, (x^L_{ju}, x^U_{ju})\right)$ *is a TIVFN for* $j = 1, 2, \ldots, n$. *According to Equations (11) and (14),*

$$\mu_{P*}(X_j, X') = \frac{(x^U_{jl} - X'^U_u) + (x^L_{jl} - X'^L_u) + 4(x_{jh} - X'_h) + (x^L_{ju} - X'^L_l) + (x^U_{ju} - X'^U_l)}{2}$$

(15)

*for* $j = 1, 2, \ldots, n$.

Through the EFPR $P*$, $\mu_{P*}(X_j, X') < 0$ denotes that $X_j$ is smaller than $X'$, $\mu_{P*}(X_j, X') > 0$ indicates that $X_j$ is larger than $X'$, or $\mu_{P*}(X_j, X') = 0$ shows that $X_j$ is equal to $X'$. Since $P*$ is a total ordering relation [34] on IVFNs, $X_1, X_2, \ldots, X_n$ in $S$ are ranked based on $\mu_{P*}(X_1, X'), \mu_{P*}(X_2, X'), \ldots, \mu_{P*}(X_n, X')$.

## 3. Extending QFD and TOPSIS under IVFE

In this section, QFD and TOPSIS are generalized under IVFE for IVFMCDM with DEC. To describe IVFMCDM clearly, related computations are presented as follows.

As QFD are combined with TOPSIS for crisp values to solve MCDM problems with DEC, the computation flowchart is expressed in Figure 4.

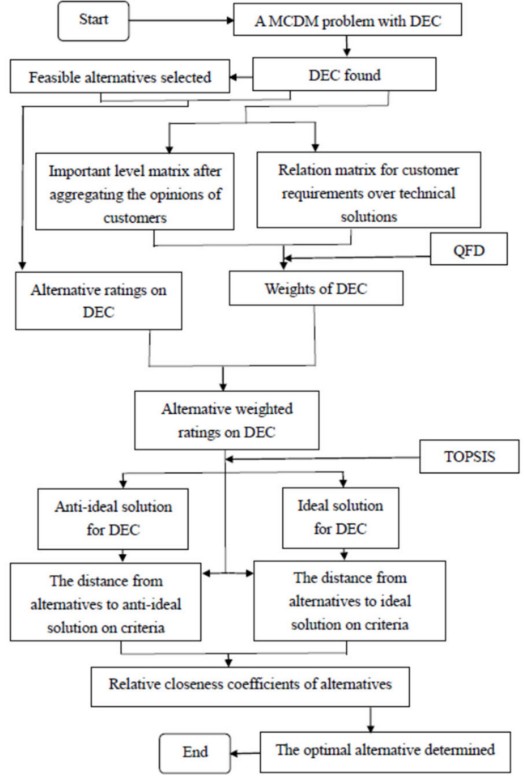

**Figure 4.** The computation flowchart of combined QFD with TOPSIS for crisp values.

Through the computation flowchart of Figure 4, related computation steps about QFD and TOPSIS extended under IVFE for IVFMCDM with DEC needs to be improved. For an IVFMCDM problem, let $C_1, C_2, \ldots, C_n$ be DEC and $W_1, W_2, \ldots, W_n$ be related weights of these DEC. These weights of DEC are derived by extending QFD under IVFE for IVFMCDM. Assume that $D_1, D_2, \ldots, D_t$ are customer requirements employed by $s$ users, and $WD_{kq}$ is a fuzzy important level of $D_k$ evaluated by the $q$th user, where $k = 1, 2, \ldots, t; q = 1, 2, \ldots, s$. $WD_1, WD_2, \ldots, WD_t$ are fuzzy important levels of $D_1, D_2, \ldots, D_t$ after aggregating the opinions of $s$ users. Therefore,

$$WD_k = \frac{1}{s} \otimes (WD_{k1} \oplus WD_{k2} \oplus \ldots \oplus WD_{ks}), \; k = 1, 2, \ldots, t \tag{16}$$

In addition,

$$WD = [WD_1, WD_2, \ldots, WD_t] \tag{17}$$

is a fuzzy important level matrix consisting of $WD_1, WD_2, \ldots, WD_t$.

According to QFD, weights $W_1, W_2, \ldots, W_n$ of DEC are obtained by merging the important level matrix above and following a relation matrix. The relation matrix between customer requirements over technical solutions is assessed by $f$ professionals. $R_{kjv}$ evaluated by the $v$th professional is the fuzzy relationship strength rating for $D_k$ over $C_j$ (i.e., technical solution), and $R_{kj}$ through $f$ professionals denotes the mean of relationship strength rating for $D_k$ over $C_j$, where $k = 1, 2, \ldots, t; j = 1, 2, \ldots, n; v = 1, 2, \ldots, f$. Hence,

$$R_{kj} = \frac{1}{f} \otimes (R_{kj1} \oplus R_{kj2} \oplus \ldots \oplus R_{kjf}), \; k = 1, 2, \ldots, t; \; j = 1, 2, \ldots, n \tag{18}$$

Then, $R$ is assumed to be the relation matrix with $R_{11}, R_{12}, \ldots, R_{tn}$ for customer requirements over technical solutions after aggregating the viewpoints of $f$ professionals, i.e.,

$$C_1 \; C_2 \; \ldots \; C_n R = \begin{array}{c} D_1 \\ D_2 \\ \vdots \\ D_t \end{array} \begin{bmatrix} R_{11} & R_{12} & \cdots & R_{1n} \\ R_{21} & R_{22} & \cdots & R_{2n} \\ \vdots & \vdots & \cdots & \vdots \\ R_{t1} & R_{t2} & \cdots & R_{tn} \end{bmatrix} \tag{19}$$

Through Equations (17) and (19), $W$ is yielded by extending QFD under fuzzy environment into a weight matrix with $W_1, W_2, \ldots, W_n$, where:

$$W_j = \frac{1}{t}[WD_1, WD_2, \ldots, WD_t] \begin{bmatrix} R_{1j} \\ R_{2j} \\ \vdots \\ R_{tj} \end{bmatrix} \tag{20}$$

$$= \frac{1}{t} \otimes (WD_1 \otimes R_{1j} \oplus WD_2 \otimes R_{2j} \oplus \ldots \oplus WD_t \otimes R_{tj})$$

for $j = 1, 2, \ldots, n$.

In addition, two fuzzy numbers [34,41] are multiplied into a PFN presented in Equations (19) and (20). Several PFNs aggregated are commonly important in extending QFD under fuzzy environment because the computations of PFNs are complex, especially for deriving IVFNs into PFNs. For instance, $WD_k = ((wd_{kl}^U, wd_{kl}^L), wd_{kh}, (wd_{ku}^L, wd_{ku}^U))$ and $R_{kj} = ((r_{kjl}^U, r_{kjl}^L), r_{kjh}, (r_{kju}^L, r_{kju}^U))$ are two TIVFNs. By extension principle,

$$WD_k \otimes R_{kj} = ((P_{kj}^U, P_{kj}^L), Q_{kj}, (Z_{kj}^L, Z_{kj}^U); (F_{kj}^L, F_{kj}^U), (T_{kj}^L, T_{kj}^U); (Y_{kj}^L, Y_{kj}^U), (V_{kj}^L, V_{kj}^U)) \tag{21}$$

for $k = 1, 2, \ldots, t; j = 1, 2, \ldots, n$.

Herein, the membership function $\mu_{WD_k \otimes R_{kj}}(x)$ of $WD_k \otimes R_{kj}$ [34] is presented to be:

$$\mu_{WD_k \otimes R_{kj}}(x) = \begin{cases} \dfrac{\left\{ -F_{kj}^U + \left[ \left( F_{kj}^U \right)^2 - 4 \left( P_{kj}^U - x \right) T_{kj}^U \right]^{1/2} \right\}}{2 T_{kj}^U} & if P_{kj}^U \leq x \leq Q_{kj} \\[4mm] \dfrac{\left\{ -F_{kj}^L + \left[ \left( F_{kj}^L \right)^2 - 4 \left( P_{kj}^L - x \right) T_{kj}^L \right]^{1/2} \right\}}{2 T_{kj}^L} & if P_{kj}^L \leq x \leq Q_{kj} \\[4mm] 1 & if x = Q_{kj} \\[2mm] \dfrac{\left\{ Y_{kj}^L - \left[ \left( Y_{kj}^L \right)^2 - 4 \left( Z_{kj}^L - x \right) V_{kj}^L \right]^{1/2} \right\}}{2 V_{kj}^L} & if Q_{kj} \leq x \leq Z_{kj}^L \\[4mm] \dfrac{\left\{ Y_{kj}^U - \left[ \left( Y_{kj}^U \right)^2 - 4 \left( Z_{kj}^U - x \right) V_{kj}^U \right]^{1/2} \right\}}{2 V_{kj}^U} & if Q_{kj} \leq x \leq Z_{kj}^U \\[4mm] 0 & otherwise \end{cases} \tag{22}$$

where

$$P_{kj}^L = wd_{kl}^L r_{kjl}^L, \; P_{kj}^U = wd_{kl}^U r_{kjl}^U,$$
$$Q_{kj} = wd_{kh} r_{kjh},$$
$$Z_{kj}^L = wd_{ku}^L r_{kju}^L, \; Z_{kj}^U = wd_{ku}^U r_{kju}^U,$$
$$F_{kj}^L = wd_{kl}^L (r_{kjh} - r_{kjl}^L) + r_{kjl}^L (wd_{kh} - wd_{kl}^L),$$
$$F_{kj}^U = wd_{kl}^U (r_{kjh} - r_{kjl}^U) + r_{kjl}^U (wd_{kh} - wd_{kl}^U),$$
$$T_{kj}^L = (wd_{kh} - wd_{kl}^L)(r_{kjh} - r_{kjl}^L), \; T_{kj}^U = (wd_{kh} - wd_{kl}^U)(r_{kjh} - r_{kjl}^U),$$
$$Y_{kj}^L = wd_{ku}^L (r_{kju}^L - r_{kjh}) + r_{kju}^L (wd_{ku}^L - wd_{kh}),$$
$$Y_{kj}^U = wd_{ku}^U (r_{kju}^U - r_{kjh}) + r_{kju}^U (wd_{ku}^U - wd_{kh}),$$
$$V_{kj}^L = (wd_{ku}^L - wd_{kh})(r_{kju}^L - r_{kjh}), \; V_{kj}^U = (wd_{ku}^U - wd_{kh})(r_{kju}^U - r_{kjh}).$$

Evidently, the multiplication of $WD_k$ and $R_{kj}$(i.e., $WD_k \otimes R_{kj}$) is a PFN, not a TIVFN. Generally, the follow-up computations of PFNs in FMCDM are too hard to derive. To solve the hard problem of deriving PFNs, the EFPR values of the fuzzy important levels over the comparison basis replace the fuzzy important levels. Owing to $WD_k(k = 1, 2, \ldots, t)$ being larger than 0, the comparison basis of these IVFNs can be assumed as 0. Then,

$$\mu_{P*}(WD_k, 0) = \frac{wd_{kl}^U + wd_{kl}^L + 4wd_{kh} + wd_{ku}^L + wd_{ku}^U}{2} \tag{23}$$

is derived by Equation (15) to denote the preference degree of $WD_k$ over 0, where $k = 1, 2, \ldots, t$. According to $\mu_{P*}(WD_k, 0)$, the matrix $W\prime$ having adjusted fuzzy weights $W_1\prime, W_2\prime, \ldots, W_n\prime$ for DEC is yielded, where:

$$W_j\prime = \frac{1}{t} \otimes [\mu_{P*}(WD_1, 0), \mu_{P*}(WD_2, 0), \ldots, \mu_{P*}(WD_t, 0)] \begin{bmatrix} R_{1j} \\ R_{2j} \\ \vdots \\ R_{tj} \end{bmatrix} = \tag{24}$$

$$\frac{1}{t} \otimes (\mu_{P*}(WD_1, 0) \otimes R_{1j} \oplus \mu_{P*}(WD_2, 0) \otimes R_{2j} \oplus \ldots \oplus \mu_{P*}(WD_t, 0) \otimes R_{tj})$$

for $j = 1, 2, \ldots, n$.

$W_j\prime$ is a TIVFN because $\mu_{P*}(WD_k, 0)$ is yielded as a crisp value and $R_{kj}$ is a TIVFN, where $k = 1, 2, \ldots, t$; $j = 1, 2, \ldots, n$. $W_1\prime, W_2\prime, \ldots, W_n\prime$ are the representation values of $W_1, W_2, \ldots, W_n$ and used in IVFMCDM with DEC.

After the adjusted fuzzy weights are derived by extending QFD under IVFE, $G_{ijv}$ is the evaluation rating assessed by the $v$th professional for $A_i$ on $C_j$, and $G_{ij}$ is the mean rating of $A_i$ on $C_j$, where $i = 1, 2, \ldots, m$; $j = 1, 2, \ldots, n$; $v = 1, 2, \ldots, f$. Therefore,

$$G_{ij} = \frac{1}{f} \otimes (G_{ij1} \oplus G_{ij2} \oplus \ldots \oplus G_{ijf}), \ i = 1, 2, \ldots, m; \ j = 1, 2, \ldots, n \tag{25}$$

Moreover, $\widetilde{G}_{ij}$ represents the normalization of $G_{ij} = \left( (g_{ijl}^U, g_{ijl}^L), g_{ijh}, (g_{iju}^L, g_{iju}^U) \right)$. The normalized computation is divided into three different statuses.

1.  $\widetilde{G}_{ij} = G_{ij}$ as $G_{ij}$ is yielded according to linguistic variables [9,10] and transformed into an IVFN in the interval [0,1];

2.  $\widetilde{G}_{ij} = ((\frac{g_{jl}^{U-}}{g_{iju}^U}, \frac{g_{jl}^{U-}}{g_{iju}^L}), \frac{g_{jl}^{U-}}{g_{ijh}}, (\frac{g_{jl}^{U-}}{g_{ijl}^L}, \frac{g_{jl}^{U-}}{g_{ijl}^U}))$ as $G_{ij}$ is assessed on cost criteria, where

    $g_{jl}^{U-} = \min\limits_{i=1,2,\ldots,m} \left\{ g_{ijl}^U \right\}, \forall j;$

3.  $\widetilde{G}_{ij} = ((\frac{g_{ijl}^U}{g_{ju}^{U+}}, \frac{g_{ijl}^L}{g_{ju}^U}), \frac{g_{ijh}}{g_{ju}^{U+}}, (\frac{g_{iju}^L}{g_{ju}^{U+}}, \frac{g_{iju}^U}{g_{ju}^U}))$ as $G_{ij}$ is evaluated on benefit criteria, where

    $g_{ju}^{U+} = \max\limits_{i=1,2,\ldots,m} \left\{ g_{iju}^U \right\}, \forall j.$

Let $\widetilde{G}_{ij} = \left( (\tilde{g}_{ijl}^U, \tilde{g}_{ijl}^L), \tilde{g}_{ijh}, (\tilde{g}_{iju}^L, \tilde{g}_{iju}^U) \right)$ be the normalized rating of the $i$th alternative on the $j$th criterion, where $i = 1, 2, \ldots, m$; $j = 1, 2, \ldots, n$. Then, the normalized rating matrix $A_i$ is presented as

$$A_i = [\widetilde{G}_{i1}, \widetilde{G}_{i2}, \ldots, \widetilde{G}_{in}], \ i = 1, 2, \ldots, m \tag{26}$$

The anti-ideal solution [29], $A^-$, found from the $m$ normalized alternatives on $n$ criteria, is:

$$A^- = [\widetilde{G}_1^-, \widetilde{G}_2^-, \ldots, \widetilde{G}_n^-], \tag{27}$$

where

$$\widetilde{G}_j^- = \left( (\tilde{g}_{jl}^{U-}, \tilde{g}_{jl}^{L-}), \tilde{g}_{jh}^-, (\tilde{g}_{ju}^{L-}, \tilde{g}_{ju}^{U-}) \right)$$
$$= (( \min\limits_{i=1,2,\ldots,m} \left\{ \tilde{g}_{ijl}^U \right\}, \min\limits_{i=1,2,\ldots,m} \left\{ \tilde{g}_{ijl}^L \right\}), \min\limits_{i=1,2,\ldots,m} \left\{ \tilde{g}_{ijh} \right\}, ( \min\limits_{i=1,2,\ldots,m} \left\{ \tilde{g}_{iju}^L \right\}, \min\limits_{i=1,2,\ldots,m} \left\{ \tilde{g}_{iju}^U \right\}))$$

for $j = 1, 2, \ldots, n$, whereas the ideal solution [29], $A^+$, found from the $m$ normalized alternatives on $n$ criteria, is:

$$A^+ = [\widetilde{G}_1^+, \widetilde{G}_2^+, \ldots, \widetilde{G}_n^+] \tag{28}$$

where

$$\widetilde{G}_j^+ = \left( (\tilde{g}_{jl}^{U+}, \tilde{g}_{jl}^{L+}), \tilde{g}_{jh}^+, (\tilde{g}_{ju}^{L+}, \tilde{g}_{ju}^{U+}) \right)$$
$$= (( \max\limits_{i=1,2,\ldots,m} \left\{ \tilde{g}_{ijl}^U \right\}, \max\limits_{i=1,2,\ldots,m} \left\{ \tilde{g}_{ijl}^L \right\}), \max\limits_{i=1,2,\ldots,m} \left\{ \tilde{g}_{ijh} \right\}, ( \max\limits_{i=1,2,\ldots,m} \left\{ \tilde{g}_{iju}^L \right\}, \max\limits_{i=1,2,\ldots,m} \left\{ \tilde{g}_{iju}^U \right\}))$$

for $j = 1, 2, \ldots, n$. To the $j$th criterion,

$$\mu_{P*}(\widetilde{G}_{ij}, \widetilde{G}_j^-) = \frac{(\tilde{g}_{ijl}^U - \tilde{g}_{ju}^{U-}) + (\tilde{g}_{ijl}^L - \tilde{g}_{ju}^{L-}) + 4(\tilde{g}_{ijh} - \tilde{g}_{jh}^-) + (\tilde{g}_{iju}^L - \tilde{g}_{jl}^{L-}) + (\tilde{g}_{iju}^U - \tilde{g}_{jl}^{U-})}{2}, \tag{29}$$

whereas

$$\mu_{P*}(\widetilde{G}_j^+, \widetilde{G}_{ij}) = \frac{(\widetilde{g}_{jl}^{U+} - \widetilde{g}_{iju}^{U}) + (\widetilde{g}_{jl}^{L+} - \widetilde{g}_{iju}^{L}) + 4(\widetilde{g}_{jh}^{+} - \widetilde{g}_{ijh}) + (\widetilde{g}_{ju}^{L+} - \widetilde{g}_{ijl}^{L}) + (\widetilde{g}_{ju}^{U+} - \widetilde{g}_{ijl}^{U})}{2},\tag{30}$$

where $i = 1, 2, \ldots, m; j = 1, 2, \ldots, n$.

According to $\mu_{P*}(\widetilde{G}_j^+, \widetilde{G}_{ij})$, $\mu_{P*}(\widetilde{G}_{ij}, \widetilde{G}_j^-)$ and $W_j'$, $D_i^-$ is calculated to indicate the weighted preference degree of $A_i$ over $A^-$, whereas $D_i^+$ is calculated to denote the weighted preference degree of $A^+$ over $A_i$ for $i = 1, 2, \ldots, m; j = 1, 2, \ldots, n$. Define

$$D_i^- = [\mu_{P*}(\widetilde{G}_{i1}, \widetilde{G}_1^-), \mu_{P*}(\widetilde{G}_{i2}, \widetilde{G}_2^-), \ldots, \mu_{P*}(\widetilde{G}_{in}, \widetilde{G}_n^-)] \begin{bmatrix} W_1' \\ W_2' \\ \vdots \\ W_n' \end{bmatrix}$$

$$= (\mu_{P*}(\widetilde{G}_{i1}, \widetilde{G}_1^-) \otimes W_1') \oplus (\mu_{P*}(\widetilde{G}_{i2}, \widetilde{G}_2^-) \otimes W_2') \oplus \ldots \oplus (\mu_{P*}(\widetilde{G}_{in}, \widetilde{G}_n^-) \otimes W_n'),\tag{31}$$

whereas

$$D_i^+ = [\mu_{P*}(\widetilde{G}_1^+, \widetilde{G}_{i1}), \mu_{P*}(\widetilde{G}_2^+, \widetilde{G}_{i2}), \ldots, \mu_{P*}(\widetilde{G}_n^+, \widetilde{G}_{in})] \begin{bmatrix} W_1' \\ W_2' \\ \vdots \\ W_n' \end{bmatrix}$$

$$= (\mu_{P*}(\widetilde{G}_1^+, \widetilde{G}_{i1}) \otimes W_1') \oplus (\mu_{P*}(\widetilde{G}_2^+, \widetilde{G}_{i2}) \otimes W_2') \oplus \ldots \oplus (\mu_{P*}(\widetilde{G}_n^+, \widetilde{G}_{in}) \otimes W_n'),\tag{32}$$

where $i = 1, 2, \ldots, m$.

According to above, $D_i^-$ and $D_i^+$ being TIVFNs are not less than 0 for $i = 1, 2, \ldots, m$. Let $D_i^- = ((d_{il}^{U-}, d_{il}^{L-}), d_{ih}^-, (d_{iu}^{L-}, d_{iu}^{U-}))$ and $D_i^+ = ((d_{il}^{U+}, d_{il}^{L+}), d_{ih}^+, (d_{iu}^{L+}, d_{iu}^{U+}))$, where $i = 1, 2, \ldots, m$. Eventually, the relative closeness coefficient $D_i$ of $A_i$ derived by EFPR is

$$D_i = \frac{\mu_{P*}(D_i^-, 0)}{\mu_{P*}(D_i^-, 0) + \mu_{P*}(D_i^+, 0)}$$

$$= \frac{d_{il}^{U-} + d_{il}^{L-} + 4d_{ih}^- + d_{iu}^{L-} + d_{iu}^{U-}}{d_{il}^{U-} + d_{il}^{L-} + 4d_{ih}^- + d_{iu}^{L-} + d_{iu}^{U-} + d_{il}^{U+} + d_{il}^{L+} + 4d_{ih}^+ + d_{iu}^{L+} + d_{iu}^{U+}},\tag{33}$$

where $\mu_{P*}(D_i^-, 0) = \dfrac{d_{il}^{U-} + d_{il}^{L-} + 4d_{ih}^- + d_{iu}^{L-} + d_{iu}^{U-}}{2}$ and $\mu_{P*}(D_i^+, 0) = \dfrac{d_{il}^{U+} + d_{il}^{L+} + 4d_{ih}^+ + d_{iu}^{L+} + d_{iu}^{U+}}{2}$ for $i = 1, 2, \ldots, m$. Obviously, the larger the value of $D_i$ is, the closer the ideal solution is. On the contrary, the lesser the value of $D_i$ is, the closer the anti-ideal solution is. The value of $D_i$ in the optimal alternative is farther from 0 and approaches 1 than others. In other words, these alternatives are ranked through $D_1, D_2, \ldots, D_m$. As $D_1, D_2, \ldots, D_m$ are computed, the ranking order for the $m$ alternatives is correspondingly determined. The best alternative will be found and IVFMCDM with DEC is finished.

To sum up, QFD and TOPSIS extended under IVFE for IVFMCDM with DEC can be summarized in Figure 5. Through the comparison between Figures 4 and 5, EFPR is critical and important for fuzzy extension of QFD and TOPSIS in IVFMCDM.

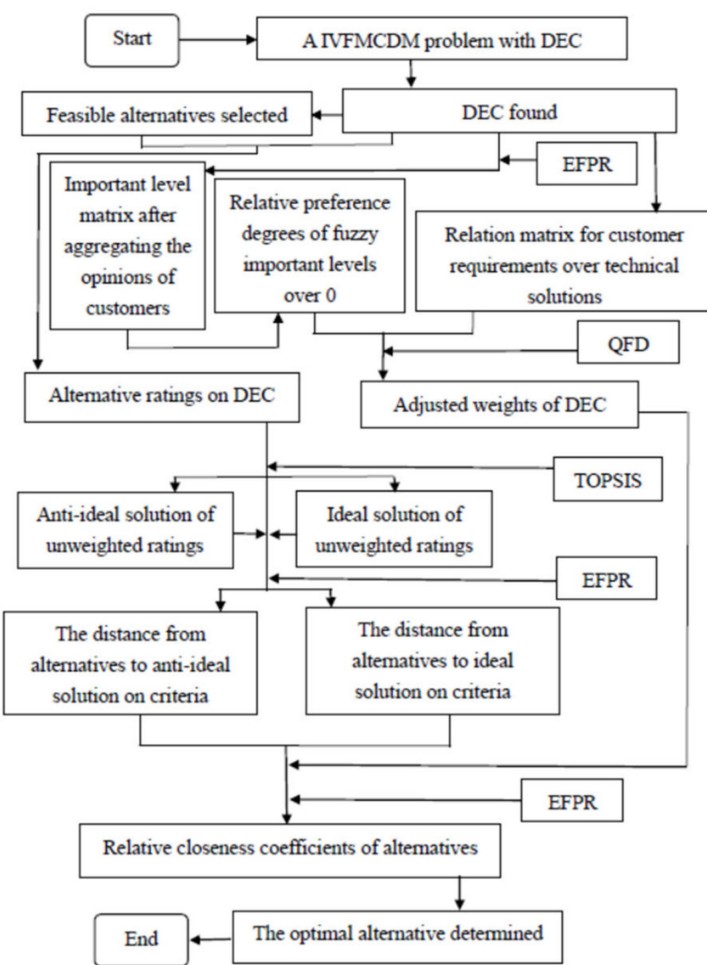

**Figure 5.** The computation flowchart of combined QFD with TOPSIS for IVFNs.

## 4. A Numerical Example about Service Performance Evaluation of International Container Ports with DEC

A numerical example similar to Wang's [29] approach is illustrated to demonstrate the IVFMCDM mentioned above clearly. In the illustrated example, 12 international container ports described in Section 1 are measured their service performance through customers based on DEC. To the IVFMCDM problem with DEC, the 12 ports, denoted by A1, A2, . . . , A12, are evaluated based on the 13 evaluation criteria, C1, C2, . . . , C13, that are also described in Section 1. In addition, convenience (D1), efficiency (D2), and safety or security (D3) employed by customers denote three customer requirements, and WD1, WD2, and WD3 indicate the fuzzy importance levels of D1, D2, and D3. These customers measure importance levels of D1, D2, and D3 by linguistic variables. The linguistic variables and their corresponding fuzzy numbers are shown in Table 1, and related assessments displayed by users are expressed in Table 2. Then, fuzzy importance levels of customer requirements based on data of Tables 1 and 2 are aggregated in Table 3.

**Table 1.** Linguistic variables and corresponding fuzzy numbers.

| Linguistic Variables | Fuzzy Numbers |
|---|---|
| Very low (VL) | ((0,0),0,(0.1,0.2)) |
| Low (L) | ((0.1,0.2)),0.3,(0.4,0.5)) |
| Medium (M) | ((0.3,0.4),0.5,(0.6,0.7)) |
| High (H) | ((0.5,0.6),0.7,(0.8,0.9)) |
| Very high (VH) | ((0.8,0.9),1,(1,1)) |

**Table 2.** Linguistic assessments for customer requirements.

| Customer Requirements | Assessments | | | | |
|---|---|---|---|---|---|
| | VL | L | M | H | VH |
| D1 | 5 | 9 | 13 | 26 | 22 |
| D2 | 12 | 10 | 18 | 19 | 16 |
| D3 | 13 | 9 | 9 | 19 | 25 |

**Table 3.** Fuzzy importance levels for customer requirements.

| Customer Requirements | Importance Levels |
|---|---|
| D1 | ((0.472,0.565),0.659,(0.729,0.800)) |
| D2 | ((0.383,0.467),0.551,(0.629,0.708)) |
| D3 | ((0.441,0.524),0.607,(0.673,0.740)) |

Using data from Table 3, the relative preference degrees of fuzzy importance levels for customer requirements over 0 calculated by Equation (15) are displayed in Table 4.

**Table 4.** Relative preference degrees of fuzzy importance levels over zero.

| Customer Requirements | Relative Preference Degrees |
|---|---|
| D1 | 2.601 |
| D2 | 2.195 |
| D3 | 2.403 |

In total, 13 technical solutions employed by experts are utilized to indicate DEC, and weights (W1, W2, ... , W13) of the evaluation criteria (C1, C2, ... , C13) are shown in the service performance evaluation problem. Moreover, linguistic relationship strength assessments for three customer requirements for technical solutions are shown in Table 5. The messages are obtained through the previous linguistic variables, too. Then, fuzzy relationship strength ratings for D1, D2, and D3 on C1, C2, ... , C13 are obtained to construct a relation matrix according to the data of Table 1 and the linguistic relationship strength ratings of Table 5. In addition, the relation matrix is presented in Table 6.

**Table 5.** Linguistic relationship strength assessments for customer requirements on technical solutions.

| | D1 | | | | | D2 | | | | | D3 | | | | |
|---|---|---|---|---|---|---|---|---|---|---|---|---|---|---|---|
| | VL | L | M | H | VH | VL | L | M | H | VH | VL | L | M | H | VH |
| C1 | 1 | 0 | 3 | 1 | 5 | 1 | 1 | 3 | 2 | 3 | 0 | 0 | 0 | 5 | 5 |
| C2 | 0 | 0 | 2 | 3 | 5 | 1 | 1 | 2 | 3 | 3 | 0 | 1 | 3 | 3 | 3 |
| C3 | 1 | 2 | 2 | 2 | 3 | 0 | 0 | 3 | 4 | 3 | 0 | 1 | 4 | 2 | 3 |
| C4 | 0 | 0 | 1 | 3 | 6 | 2 | 3 | 3 | 1 | 1 | 2 | 2 | 3 | 3 | 0 |
| C5 | 0 | 1 | 1 | 2 | 6 | 1 | 1 | 1 | 3 | 4 | 3 | 1 | 4 | 1 | 1 |
| C6 | 3 | 1 | 1 | 5 | 0 | 1 | 0 | 3 | 2 | 4 | 0 | 1 | 1 | 2 | 6 |
| C7 | 4 | 1 | 3 | 1 | 1 | 0 | 0 | 1 | 4 | 5 | 3 | 2 | 3 | 1 | 1 |
| C8 | 2 | 2 | 2 | 1 | 3 | 1 | 0 | 1 | 3 | 5 | 0 | 1 | 2 | 2 | 5 |
| C9 | 0 | 1 | 0 | 3 | 6 | 1 | 0 | 1 | 2 | 6 | 0 | 1 | 1 | 3 | 5 |
| C10 | 1 | 2 | 6 | 1 | 0 | 1 | 1 | 0 | 2 | 6 | 4 | 2 | 3 | 1 | 0 |
| C11 | 2 | 3 | 1 | 4 | 0 | 0 | 1 | 1 | 5 | 3 | 1 | 1 | 3 | 2 | 3 |
| C13 | 0 | 1 | 2 | 2 | 5 | 1 | 2 | 6 | 0 | 1 | 1 | 0 | 2 | 2 | 5 |
| C13 | 0 | 2 | 6 | 1 | 1 | 0 | 2 | 2 | 3 | 3 | 1 | 1 | 0 | 2 | 6 |

Through the data of Tables 4 and 6, an adjusted fuzzy weight matrix computed by Equation (24) for the DEC is derived in Table 7.

**Table 6.** The relation matrix for three customer requirements on 13 technical solutions.

| | **C1** | **C2** | **C3** |
|---|---|---|---|
| D1 | ((0.54,0.63),0.72,(0.77,0.82)) | ((0.61,0.71),0.81,(0.86,0.91)) | ((0.42,0.51),0.60,(0.67,0.74)) |
| D2 | ((0.44,0.53),0.62,(0.69,0.76)) | ((0.46,0.55),0.64,(0.71,0.78)) | ((0.53,0.63),0.73,(0.80,0.87)) |
| D3 | ((0.65,0.75),0.85,(0.90,0.95)) | ((0.49,0.59),0.69,(0.76,0.83)) | ((0.47,0.57),0.67,(0.74,0.81)) |
| | **C4** | **C5** | **C6** |
| D1 | ((0.66,0.76),0.86,(0.90,0.94)) | ((0.62,0.72),0.82,(0.86,0.90)) | ((0.29,0.36),0.43,(0.53,0.63)) |
| D2 | ((0.25,0.33),0.41,(0.50,0.59)) | ((0.51,0.60),0.69,(0.75,0.81)) | ((0.51,0.60),0.69,(0.75,0.81)) |
| D3 | ((0.26,0.34),0.42,(0.52,0.62)) | ((0.26,0.33),0.40,(0.49,0.58)) | ((0.62,0.72),0.82,(0.86,0.90)) |
| | **C7** | **C8** | **C9** |
| D1 | ((0.23,0.29),0.35,(0.44,0.53)) | ((0.37,0.45),0.53,(0.60,0.67)) | ((0.33,0.43),0.53,(0.62,0.71)) |
| D2 | ((0.63,0.73),0.83,(0.88,0.93)) | ((0.58,0.67),0.76,(0.81,0.86)) | ((0.47,0.57),0.67,(0.74,0.81)) |
| D3 | ((0.24,0.31),0.38,(0.47,0.56)) | ((0.57,0.67),0.77,(0.82,0.87)) | ((0.59,0.68),0.77,(0.81,0.85)) |
| | **C10** | **C11** | **C12** |
| D1 | ((0.25,0.34),0.43,(0.53,0.63)) | ((0.26,0.34),0.42,(0.52,0.62)) | ((0.57,0.67),0.77,(0.82,0.87)) |
| D2 | ((0.59,0.68),0.77,(0.81,0.85)) | ((0.53,0.63),0.73,(0.80,0.87)) | ((0.28,0.37),0.46,(0.55,0.64)) |
| D3 | ((0.16,0.22),0.28,(0.38,0.48)) | ((0.44,0.53),0.62,(0.69,0.76)) | ((0.56,0.65),0.74,(0.79,0.84)) |
| | **C13** | | |
| D1 | ((0.33,0.43),0.53,(0.62,0.71)) | | |
| D2 | ((0.47,0.57),0.67,(0.74,0.81)) | | |
| D3 | ((0.59,0.68),0.77,(0.81,0.85)) | | |

**Table 7.** The adjusted fuzzy weight matrix for DEC.

| **W1′** | **W2′** | **W3′** |
|---|---|---|
| ((1.311,1.535),1.758,(1.893,2.028)) | ((1.258,1.490),1.723,(1.874,2.024)) | ((1.128,1.360),1.591,(1.759,1.927)) |
| **W4′** | **W5′** | **W6′** |
| ((0.963,1.173),1.382,(1.562,1.743)) | ((1.119,1.327),1.536,(1.687,1.837)) | ((1.121,1.328),1.534,(1.697,1.860)) |
| **W7′** | **W8′** | **W9′** |
| ((0.852,1.034),1.215,(1.402,1.588)) | ((1.202,1.417),1.632,(1.769,1.907)) | ((1.102,1.334),1.566,(1.728,1.889)) |
| **W10′** | **W11′** | **W12′** |
| ((0.776,0.968),1.160,(1.356,1.552)) | ((0.966,1.180),1.395,(1.589,1.783)) | ((1.147,1.372),1.597,(1.746,1.895)) |
| **W13′** | | |
| ((1.102,1.334),1.566,(1.728,1.889)) | | |

As the adjusted fuzzy weight matrix is derived by extending QFD under a fuzzy environment, these above users, recognized as experts, are also employed to evaluate the service performance of 12 ports through linguistic variables of Table 1. These linguistic performance ratings utilized by professionals against evaluation criteria are presented in Table 8, and then the fuzzy performance matrix is aggregated in Table 9.

**Table 8.** Linguistic performance ratings of international container ports on evaluation criteria.

| | **A1** | | | | | **A2** | | | | | **A3** | | | | |
|---|---|---|---|---|---|---|---|---|---|---|---|---|---|---|---|
| | **VL** | **L** | **M** | **H** | **VH** | **VL** | **L** | **M** | **H** | **VH** | **VL** | **L** | **M** | **H** | **VH** |
| C1 | 0 | 3 | 3 | 1 | 3 | 1 | 1 | 1 | 4 | 3 | 1 | 4 | 4 | 0 | 1 |
| C2 | 1 | 1 | 1 | 4 | 3 | 0 | 0 | 3 | 3 | 4 | 0 | 1 | 3 | 3 | 3 |
| C3 | 2 | 1 | 3 | 1 | 3 | 2 | 2 | 3 | 0 | 3 | 2 | 0 | 2 | 0 | 6 |
| C4 | 2 | 0 | 2 | 3 | 3 | 0 | 6 | 1 | 1 | 2 | 1 | 1 | 1 | 3 | 4 |
| C5 | 3 | 1 | 4 | 1 | 1 | 1 | 1 | 2 | 3 | 3 | 0 | 1 | 3 | 1 | 5 |
| C6 | 0 | 3 | 3 | 3 | 1 | 2 | 1 | 5 | 2 | 0 | 1 | 5 | 0 | 2 | 2 |

**Table 8.** *Cont.*

| | A1 | | | | | A2 | | | | | A3 | | | |
|---|---|---|---|---|---|---|---|---|---|---|---|---|---|---|---|
| | **VL** | **L** | **M** | **H** | **VH** | **VL** | **L** | **M** | **H** | **VH** | **VL** | **L** | **M** | **H** | **VH** |
| C7 | 0 | 1 | 1 | 5 | 3 | 3 | 3 | 3 | 0 | 1 | 0 | 0 | 3 | 5 | 2 |
| C8 | 1 | 3 | 2 | 3 | 1 | 1 | 1 | 1 | 4 | 3 | 2 | 3 | 2 | 3 | 0 |
| C9 | 2 | 0 | 3 | 2 | 3 | 3 | 1 | 3 | 0 | 3 | 2 | 3 | 3 | 1 | 1 |
| C10 | 1 | 4 | 1 | 3 | 1 | 1 | 1 | 2 | 3 | 3 | 2 | 5 | 1 | 0 | 2 |
| C11 | 1 | 1 | 3 | 4 | 1 | 4 | 1 | 1 | 1 | 3 | 1 | 0 | 6 | 1 | 2 |
| C13 | 1 | 2 | 3 | 1 | 3 | 1 | 2 | 5 | 0 | 2 | 0 | 4 | 4 | 1 | 1 |
| C13 | 0 | 0 | 2 | 5 | 3 | 1 | 3 | 0 | 3 | 3 | 1 | 0 | 1 | 3 | 5 |

| | A4 | | | | | A5 | | | | | A6 | | | |
|---|---|---|---|---|---|---|---|---|---|---|---|---|---|---|---|
| | **VL** | **L** | **M** | **H** | **VH** | **VL** | **L** | **M** | **H** | **VH** | **VL** | **L** | **M** | **H** | **VH** |
| C1 | 2 | 3 | 0 | 2 | 3 | 5 | 0 | 1 | 2 | 2 | 1 | 1 | 2 | 3 | 3 |
| C2 | 0 | 0 | 4 | 0 | 6 | 2 | 1 | 5 | 0 | 2 | 1 | 3 | 1 | 2 | 3 |
| C3 | 1 | 0 | 3 | 3 | 3 | 1 | 1 | 0 | 4 | 4 | 2 | 0 | 2 | 3 | 3 |
| C4 | 1 | 1 | 0 | 4 | 4 | 2 | 1 | 5 | 2 | 0 | 1 | 6 | 1 | 2 | 0 |
| C5 | 3 | 3 | 1 | 3 | 0 | 0 | 3 | 3 | 3 | 1 | 5 | 1 | 2 | 0 | 2 |
| C6 | 2 | 3 | 2 | 0 | 3 | 1 | 3 | 3 | 0 | 3 | 2 | 0 | 4 | 2 | 2 |
| C7 | 1 | 1 | 6 | 1 | 1 | 2 | 0 | 4 | 3 | 1 | 3 | 0 | 2 | 2 | 3 |
| C8 | 0 | 1 | 1 | 5 | 3 | 5 | 0 | 2 | 3 | 0 | 1 | 1 | 5 | 1 | 2 |
| C9 | 3 | 3 | 4 | 0 | 0 | 2 | 2 | 2 | 0 | 4 | 2 | 2 | 0 | 6 | 0 |
| C10 | 0 | 4 | 4 | 1 | 1 | 2 | 3 | 3 | 1 | 1 | 1 | 0 | 6 | 3 | 0 |
| C11 | 2 | 2 | 0 | 3 | 3 | 1 | 1 | 2 | 3 | 3 | 1 | 1 | 1 | 1 | 6 |
| C13 | 1 | 1 | 1 | 4 | 3 | 1 | 0 | 1 | 3 | 5 | 2 | 0 | 3 | 0 | 5 |
| C13 | 2 | 0 | 3 | 2 | 3 | 0 | 0 | 2 | 6 | 2 | 3 | 3 | 3 | 0 | 1 |

| | A7 | | | | | A8 | | | | | A9 | | | |
|---|---|---|---|---|---|---|---|---|---|---|---|---|---|---|---|
| | **VL** | **L** | **M** | **H** | **VH** | **VL** | **L** | **M** | **H** | **VH** | **VL** | **L** | **M** | **H** | **VH** |
| C1 | 2 | 0 | 2 | 3 | 3 | 3 | 3 | 0 | 3 | 1 | 1 | 0 | 6 | 3 | 0 |
| C2 | 0 | 2 | 5 | 1 | 2 | 3 | 1 | 3 | 0 | 3 | 2 | 2 | 2 | 3 | 1 |
| C3 | 3 | 0 | 3 | 0 | 4 | 1 | 1 | 6 | 1 | 1 | 2 | 1 | 4 | 1 | 2 |
| C4 | 0 | 1 | 3 | 3 | 3 | 5 | 0 | 1 | 2 | 2 | 3 | 3 | 3 | 1 | 0 |
| C5 | 0 | 2 | 2 | 6 | 0 | 1 | 1 | 0 | 6 | 2 | 5 | 0 | 1 | 2 | 2 |
| C6 | 1 | 0 | 1 | 2 | 6 | 0 | 1 | 1 | 5 | 3 | 0 | 0 | 4 | 3 | 3 |
| C7 | 3 | 3 | 1 | 3 | 0 | 3 | 0 | 6 | 0 | 1 | 1 | 1 | 4 | 4 | 0 |
| C8 | 1 | 1 | 1 | 1 | 6 | 6 | 1 | 1 | 1 | 1 | 2 | 1 | 5 | 0 | 2 |
| C9 | 2 | 3 | 3 | 0 | 2 | 1 | 1 | 3 | 2 | 3 | 2 | 0 | 2 | 2 | 4 |
| C10 | 3 | 3 | 3 | 0 | 1 | 2 | 3 | 2 | 0 | 3 | 1 | 3 | 3 | 3 | 0 |
| C11 | 1 | 1 | 4 | 3 | 1 | 1 | 1 | 4 | 1 | 3 | 3 | 3 | 0 | 0 | 4 |
| C13 | 1 | 0 | 2 | 5 | 2 | 0 | 3 | 0 | 5 | 2 | 0 | 0 | 3 | 5 | 2 |
| C13 | 0 | 4 | 4 | 1 | 1 | 0 | 4 | 1 | 4 | 1 | 0 | 4 | 3 | 3 | 0 |

| | A10 | | | | | A11 | | | | | A12 | | | |
|---|---|---|---|---|---|---|---|---|---|---|---|---|---|---|---|
| | **VL** | **L** | **M** | **H** | **VH** | **VL** | **L** | **M** | **H** | **VH** | **VL** | **L** | **M** | **H** | **VH** |
| C1 | 2 | 0 | 5 | 0 | 3 | 1 | 1 | 1 | 1 | 6 | 1 | 1 | 6 | 2 | 0 |
| C2 | 0 | 5 | 5 | 0 | 0 | 6 | 0 | 4 | 0 | 0 | 6 | 1 | 1 | 1 | 1 |
| C3 | 0 | 1 | 1 | 3 | 5 | 5 | 0 | 1 | 2 | 2 | 3 | 3 | 3 | 1 | 0 |
| C4 | 1 | 0 | 1 | 4 | 4 | 0 | 0 | 3 | 3 | 4 | 1 | 0 | 1 | 2 | 6 |
| C5 | 6 | 3 | 1 | 0 | 0 | 2 | 3 | 1 | 3 | 1 | 2 | 2 | 2 | 3 | 1 |
| C6 | 3 | 0 | 6 | 0 | 1 | 1 | 1 | 1 | 3 | 4 | 2 | 0 | 3 | 2 | 3 |
| C7 | 0 | 0 | 4 | 3 | 3 | 5 | 0 | 1 | 1 | 3 | 5 | 0 | 5 | 0 | 0 |
| C8 | 4 | 3 | 1 | 1 | 1 | 0 | 4 | 2 | 4 | 0 | 1 | 1 | 1 | 1 | 6 |
| C9 | 0 | 2 | 2 | 6 | 0 | 1 | 1 | 4 | 4 | 0 | 3 | 5 | 2 | 0 | 0 |
| C10 | 0 | 3 | 4 | 3 | 0 | 2 | 3 | 3 | 2 | 0 | 2 | 3 | 0 | 4 | 1 |
| C11 | 6 | 0 | 2 | 2 | 0 | 2 | 0 | 2 | 2 | 4 | 0 | 0 | 0 | 5 | 5 |
| C13 | 2 | 1 | 5 | 0 | 2 | 0 | 0 | 0 | 4 | 6 | 2 | 3 | 3 | 0 | 2 |
| C13 | 0 | 0 | 6 | 2 | 2 | 0 | 0 | 2 | 3 | 5 | 0 | 1 | 3 | 3 | 3 |

**Table 9.** Fuzzy performance matrix for international container ports on criteria.

|  | C1 | C2 | C3 |
|---|---|---|---|
| A1 | ((0.41,0.51),0.61,(0.68,0.75)) | ((0.48,0.57),0.66,(0.73,0.80)) | ((0.39,0.47),0.55,(0.62,0.69)) |
| A2 | ((0.48,0.57),0.66,(0.73,0.80)) | ((0.56,0.66),0.76,(0.82,0.88)) | ((0.35,0.43),0.51,(0.58,0.65)) |
| A3 | ((0.24,0.33),0.42,(0.51,0.60)) | ((0.49,0.59),0.69,(0.76,0.83)) | ((0.54,0.62),0.70,(0.74,0.78)) |
| A4 | ((0.37,0.45),0.53,(0.60,0.67)) | ((0.60,0.70),0.80,(0.84,0.88)) | ((0.48,0.57),0.66,(0.73,0.80)) |
| A5 | ((0.29,0.34),0.39,(0.47,0.55)) | ((0.32,0.40),0.48,(0.56,0.64)) | ((0.53,0.62),0.71,(0.77,0.83)) |
| A6 | ((0.46,0.55),0.64,(0.71,0.78)) | ((0.40,0.49),0.58,(0.65,0.72)) | ((0.45,0.53),0.61,(0.68,0.75)) |
| A7 | ((0.45,0.53),0.61,(0.68,0.75)) | ((0.38,0.48),0.58,(0.66,0.74)) | ((0.41,0.48),0.55,(0.61,0.67)) |
| A8 | ((0.26,0.33),0.40,(0.49,0.58)) | ((0.34,0.41),0.48,(0.55,0.62)) | ((0.32,0.41),0.50,(0.59,0.68)) |
| A9 | ((0.33,0.42),0.51,(0.61,0.71)) | ((0.31,0.39),0.47,(0.56,0.65)) | ((0.34,0.42),0.50,(0.58,0.66)) |
| A10 | ((0.39,0.47),0.55,(0.62,0.69)) | ((0.20,0.30),0.40,(0.50,0.60)) | ((0.59,0.69),0.79,(0.84,0.89)) |
| A11 | ((0.57,0.66),0.75,(0.79,0.83)) | ((0.12,0.16),0.20,(0.30,0.40)) | ((0.29,0.34),0.39,(0.47,0.55)) |
| A12 | ((0.29,0.38),0.47,(0.57,0.67)) | ((0.17,0.21),0.25,(0.34,0.43)) | ((0.17,0.24),0.31,(0.41,0.51)) |

|  | C4 | C5 | C6 |
|---|---|---|---|
| A1 | ((0.45,0.53),0.61,(0.68,0.75)) | ((0.26,0.33),0.40,(0.49,0.58)) | ((0.35,0.45),0.55,(0.64,0.73)) |
| A2 | ((0.30,0.40),0.50,(0.58,0.66)) | ((0.46,0.55),0.64,(0.71,0.78)) | ((0.26,0.34),0.42,(0.52,0.62)) |
| A3 | ((0.51,0.60),0.69,(0.75,0.81)) | ((0.55,0.65),0.75,(0.80,0.85)) | ((0.31,0.40),0.49,(0.57,0.65)) |
| A4 | ((0.53,0.62),0.71,(0.77,0.83)) | ((0.21,0.28),0.35,(0.45,0.55)) | ((0.33,0.41),0.49,(0.56,0.63)) |
| A5 | ((0.26,0.34),0.42,(0.52,0.62)) | ((0.35,0.45),0.55,(0.64,0.73)) | ((0.36,0.45),0.54,(0.61,0.68)) |
| A6 | ((0.19,0.28),0.37,(0.47,0.57)) | ((0.23,0.28),0.33,(0.41,0.49)) | ((0.38,0.46),0.54,(0.62,0.70)) |
| A7 | ((0.49,0.59),0.69,(0.76,0.83)) | ((0.38,0.48),0.58,(0.68,0.78)) | ((0.61,0.70),0.79,(0.83,0.87)) |
| A8 | ((0.29,0.34),0.39,(0.47,0.55)) | ((0.47,0.56),0.65,(0.73,0.81)) | ((0.53,0.63),0.73,(0.80,0.87)) |
| A9 | ((0.17,0.24),0.31,(0.41,0.51)) | ((0.29,0.34),0.39,(0.47,0.55)) | ((0.51,0.61),0.71,(0.78,0.85)) |
| A10 | ((0.55,0.64),0.73,(0.79,0.85)) | ((0.06,0.10),0.14,(0.24,0.34)) | ((0.26,0.33),0.40,(0.49,0.58)) |
| A11 | ((0.56,0.66),0.76,(0.82,0.88)) | ((0.29,0.37),0.45,(0.54,0.63)) | ((0.51,0.60),0.69,(0.75,0.81)) |
| A12 | ((0.61,0.70),0.79,(0.83,0.87)) | ((0.31,0.39),0.47,(0.56,0.65)) | ((0.43,0.51),0.59,(0.66,0.73)) |

|  | C7 | C8 | C9 |
|---|---|---|---|
| A1 | ((0.53,0.63),0.73,(0.80,0.87)) | ((0.32,0.41),0.50,(0.59,0.68)) | ((0.43,0.51),0.59,(0.66,0.73)) |
| A2 | ((0.20,0.27),0.34,(0.43,0.52)) | ((0.48,0.57),0.66,(0.73,0.80)) | ((0.34,0.41),0.48,(0.55,0.62)) |
| A3 | ((0.50,0.60),0.70,(0.78,0.86)) | ((0.24,0.32),0.40,(0.50,0.60)) | ((0.25,0.33),0.41,(0.50,0.59)) |
| A4 | ((0.32,0.41),0.50,(0.59,0.68)) | ((0.53,0.63),0.73,(0.80,0.87)) | ((0.15,0.22),0.29,(0.39,0.49)) |
| A5 | ((0.35,0.43),0.51,(0.60,0.69)) | ((0.21,0.26),0.31,(0.41,0.51)) | ((0.40,0.48),0.56,(0.62,0.68)) |
| A6 | ((0.40,0.47),0.54,(0.61,0.68)) | ((0.37,0.46),0.55,(0.63,0.71)) | ((0.32,0.40),0.48,(0.58,0.68)) |
| A7 | ((0.21,0.28),0.35,(0.45,0.55)) | ((0.57,0.66),0.75,(0.79,0.83)) | ((0.28,0.36),0.44,(0.52,0.60)) |
| A8 | ((0.26,0.33),0.40,(0.49,0.58)) | ((0.17,0.21),0.25,(0.34,0.43)) | ((0.44,0.53),0.62,(0.69,0.76)) |
| A9 | ((0.33,0.42),0.51,(0.61,0.71)) | ((0.32,0.40),0.48,(0.56,0.64)) | ((0.48,0.56),0.64,(0.70,0.76)) |
| A10 | ((0.51,0.61),0.71,(0.78,0.85)) | ((0.19,0.25),0.31,(0.40,0.49)) | ((0.38,0.48),0.58,(0.68,0.78)) |
| A11 | ((0.32,0.37),0.42,(0.49,0.56)) | ((0.30,0.40),0.50,(0.60,0.70)) | ((0.33,0.42),0.51,(0.61,0.71)) |
| A12 | ((0.15,0.20),0.25,(0.35,0.45)) | ((0.57,0.66),0.75,(0.79,0.83)) | ((0.11,0.18),0.25,(0.35,0.45)) |

|  | C10 | C11 | C12 |
|---|---|---|---|
| A1 | ((0.30,0.39),0.48,(0.57,0.66)) | ((0.38,0.47),0.56,(0.65,0.74)) | ((0.40,0.49),0.58,(0.65,0.72)) |
| A2 | ((0.46,0.55),0.64,(0.71,0.78)) | ((0.33,0.39),0.45,(0.52,0.59)) | ((0.33,0.42),0.51,(0.59,0.67)) |
| A3 | ((0.24,0.32),0.40,(0.48,0.56)) | ((0.39,0.48),0.57,(0.65,0.73)) | ((0.29,0.39),0.49,(0.58,0.67)) |
| A4 | ((0.29,0.39),0.49,(0.58,0.67)) | ((0.41,0.49),0.57,(0.64,0.71)) | ((0.48,0.57),0.66,(0.73,0.80)) |
| A5 | ((0.25,0.33),0.41,(0.50,0.59)) | ((0.46,0.55),0.64,(0.71,0.78)) | ((0.58,0.67),0.76,(0.81,0.86)) |
| A6 | ((0.33,0.42),0.51,(0.61,0.71)) | ((0.57,0.66),0.75,(0.79,0.83)) | ((0.49,0.57),0.65,(0.70,0.75)) |
| A7 | ((0.20,0.27),0.34,(0.43,0.52)) | ((0.36,0.45),0.54,(0.63,0.72)) | ((0.47,0.56),0.65,(0.73,0.81)) |
| A8 | ((0.33,0.41),0.49,(0.56,0.63)) | ((0.42,0.51),0.60,(0.67,0.74)) | ((0.44,0.54),0.64,(0.72,0.80)) |
| A9 | ((0.27,0.36),0.45,(0.55,0.65)) | ((0.35,0.42),0.49,(0.55,0.61)) | ((0.50,0.60),0.70,(0.78,0.86)) |
| A10 | ((0.30,0.40),0.50,(0.60,0.70)) | ((0.16,0.20),0.24,(0.34,0.44)) | ((0.32,0.40),0.48,(0.56,0.64)) |
| A11 | ((0.22,0.30),0.38,(0.48,0.58)) | ((0.48,0.56),0.64,(0.70,0.76)) | ((0.68,0.78),0.88,(0.92,0.96)) |
| A12 | ((0.31,0.39),0.47,(0.56,0.65)) | ((0.65,0.75),0.85,(0.90,0.95)) | ((0.28,0.36),0.44,(0.52,0.60)) |

|  | C13 |
|---|---|
| A1 | ((0.55,0.65),0.75,(0.82,0.89)) |
| A2 | ((0.42,0.51),0.60,(0.67,0.74)) |

**Table 9.** *Cont.*

| | C13 |
|---|---|
| A3 | ((0.58,0.67),0.76,(0.81,0.86)) |
| A4 | ((0.43,0.51),0.59,(0.66,0.73)) |
| A5 | ((0.52,0.62),0.72,(0.80,0.88)) |
| A6 | ((0.20,0.27),0.34,(0.43,0.52)) |
| A7 | ((0.29,0.39),0.49,(0.58,0.67)) |
| A8 | ((0.35,0.45),0.55,(0.64,0.73)) |
| A9 | ((0.28,0.38),0.48,(0.58,0.68)) |
| A10 | ((0.44,0.54),0.64,(0.72,0.80)) |
| A11 | ((0.61,0.71),0.81,(0.86,0.91)) |
| A12 | ((0.49,0.59),0.69,(0.76,0.83)) |

Based on the entries of Table 9, these alternatives on varied criteria have different strengths for ratings. Due to the ratings constructed based upon IVIFNs, it is very difficult to compare these alternatives based on criteria, and the evidence is shown in the radar chart of Figure 6. In fact, the criteria comparison complexity of these alternatives is still high and even IVIFNs are degenerated to be crisp values through the concept of mean. The situation is expressed in the radar chart of Figure 7. Therefore, aggregating alternative ratings on varied criteria is necessary and criterial.

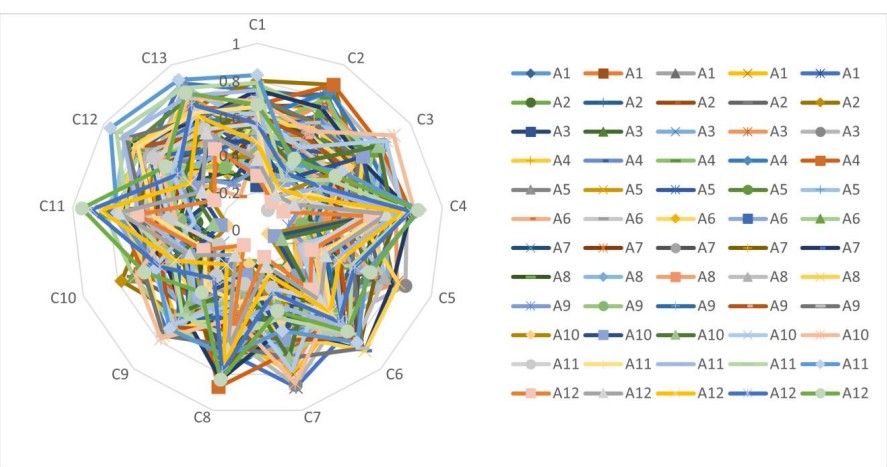

**Figure 6.** The criteria comparison of twelve alternatives composed of IVIFNs.

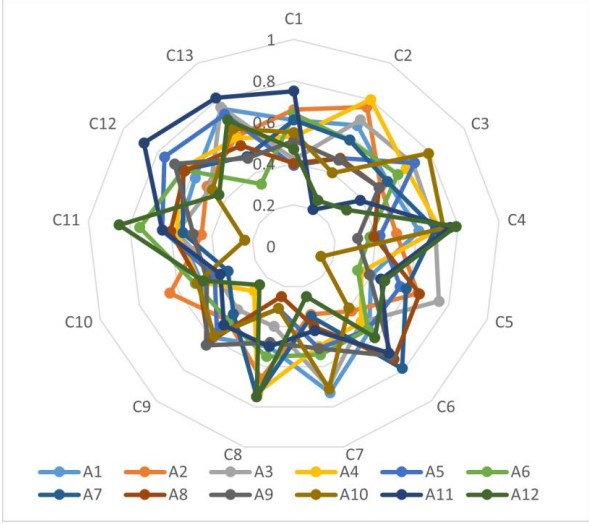

**Figure 7.** The criteria comparison of 12 alternatives composed of crisp values.

By extending TOPSIS under IVFE, the anti-ideal and ideal solutions of the 12 international container ports against 13 evaluation criteria according to the data of Table 9 are derived in Table 10.

**Table 10.** Anti-ideal and ideal solutions of twelve international container ports on all criteria.

|  | **C1** | **C2** | **C3** |
|---|---|---|---|
| Anti-ideal solution | ((0.24,0.33),0.39,(0.47,0.55)) | ((0.12,0.16),0.20,(0.30,0.40)) | ((0.17,0.24),0.31,(0.41,0.51)) |
| Ideal solution | ((0.57,0.66),0.75,(0.79,0.83)) | ((0.60,0.70),0.80,(0.84,0.88)) | ((0.59,0.69),0.79,(0.84,0.89)) |
|  | **C4** | **C5** | **C6** |
| Anti-ideal solution | ((0.17,0.24),0.31,(0.41,0.51)) | ((0.06,0.10),0.14,(0.24,0.34)) | ((0.26,0.33),0.40,(0.49,0.58)) |
| Ideal solution | ((0.61,0.70),0.79,(0.83,0.88)) | ((0.55,0.65),0.75,(0.80,0.85)) | ((0.61,0.70),0.79,(0.83,0.87)) |
|  | **C7** | **C8** | **C9** |
| Anti-ideal solution | ((0.15,0.20),0.25,(0.35,0.45)) | ((0.17,0.21),0.25,(0.34,0.43)) | ((0.11,0.18),0.25,(0.35,0.45)) |
| Ideal solution | ((0.53,0.63),0.73,(0.80,0.87)) | ((0.57,0.66),0.75,(0.80,0.87)) | ((0.48,0.56),0.64,(0.70,0.78)) |
|  | **C10** | **C11** | **C12** |
| Anti-ideal solution | ((0.20,0.27),0.34,(0.43,0.52)) | ((0.16,0.20),0.24,(0.34,0.44)) | ((0.28,0.36),0.44,(0.52,0.60)) |
| Ideal solution | ((0.46,0.55),0.64,(0.71,0.78)) | ((0.65,0.75),0.85,(0.90,0.95)) | ((0.68,0.78),0.88,(0.92,0.96)) |
|  | **C13** |  |  |
| Anti-ideal solution | ((0.20,0.27),0.34,(0.43,0.52)) |  |  |
| Ideal solution | ((0.61,0.71),0.81,(0.86,0.91)) |  |  |

Through information of Tables 7, 9 and 10, weighted preference degrees of these ports over the anti-ideal solution are derived in Table 11. Additionally, weighted preference degrees of the ideal solution over these ports are yielded in Table 12.

**Table 11.** Weighted preference degrees of international container ports over the anti-ideal solution.

|  | **Weighted Preference Degrees** |
|---|---|
| A1 | ((10.4589,12.5568),14.6547,(16.2703,17.8859)) |
| A2 | ((13.7587,16.4601),19.1614,(21.1399,23.1183)) |
| A3 | ((14.4360,17.3390),20.2419,(22.4793,24.7167)) |
| A4 | ((14.4388,17.2991),20.1595,(22.3024,24.4453)) |
| A5 | ((12.7780,15.3665),17.9550,(19.9174,21.8798)) |
| A6 | ((12.3903,14.8501),17.3099,(19.1862,21.0625)) |
| A7 | ((14.7240,17.5798),20.4355,(22.5247,24.6139)) |
| A8 | ((11.5305,13.8491),16.1677,(17.9403,19.7128)) |
| A9 | ((11.4995,13.7733),16.0472,(17.7481,19.4490)) |
| A10 | ((10.2931,12.4139),14.5347,(16.1961,17.8575)) |
| A11 | ((14.4753,17.3570),20.2388,(22.3946,24.5505)) |
| A12 | ((10.8658,13.0575),15.2492,(16.9584,18.6675)) |

**Table 12.** Weighted preference degrees of the ideal solution over international container ports.

|  | **Weighted Preference Degrees** |
|---|---|
| A1 | ((9.8545,11.7973),13.7401,(15.1954,16.6506)) |
| A2 | ((11.1390,13.3959),15.6528,(17.4328,19.2128)) |
| A3 | ((10.4617,12.5170),14.5723,(16.0934,17.6144)) |
| A4 | ((10.4589,12.5568),14.6547,(16.2703,17.8859)) |
| A5 | ((12.1197,14.4895),16.8592,(18.6553,20.4514)) |
| A6 | ((12.5074,15.0058),17.5043,(19.3865,21.2687)) |
| A7 | ((10.1737,12.2762),14.3787,(16.0480,17.7173)) |
| A8 | ((13.3672,16.0068),18.6465,(20.6324,22.6184)) |
| A9 | ((13.3982,16.0826),18.7670,(20.8246,22.8821)) |
| A10 | ((14.6046,17.4420),20.2795,(22.3766,24.4737)) |
| A11 | ((10.4224,12.4989),14.5754,(16.1781,17.7807)) |
| A12 | ((14.0450,16.8138),19.5826,(21.6332,23.6839)) |

Through entries of Tables 11 and 12, relative closeness coefficients of the 12 international container ports in service performance and their ranking order are gained in Table 13.

**Table 13.** Relative closeness coefficients and ranking order of international container ports on service performance.

|  | Relative Closeness Coefficients | Ranking Order |
|---|---|---|
| A1 | 0.5163 | 6 |
| A2 | 0.5497 | 5 |
| A3 | 0.5818 | 2 |
| A4 | 0.5788 | 4 |
| A5 | 0.5157 | 7 |
| A6 | 0.4974 | 8 |
| A7 | 0.5863 | 1 |
| A8 | 0.4645 | 9 |
| A9 | 0.4607 | 10 |
| A10 | 0.4179 | 12 |
| A11 | 0.5810 | 3 |
| A12 | 0.4383 | 11 |

The relative closeness coefficients of these ports related to service performance are, respectively, A1: 0.5163, A2: 0.5497, A3: 0.5818, A4: 0.5788, A5: 0.5157, A6: 0.4974, A7: 0.5863, A8: 0.4645, A9: 0.4607, A10: 0.4179, A11: 0.5810, and A12: 0.4383. Therefore, their ranking order determined by these relative closeness coefficients is A7 > A3 > A11 > A4 > A2 > A1 > A5 > A6 > A8 > A9 > A12 > A10 presented in the table as well. Obviously, A7 is the optimal port of the 12 international container ports, based upon service performance.

In 2016, Wang [2] evaluated these international container ports listed above based upon weakness and strength indices of FMCDM. In addition to the weakness and strength indices of FMCDM, Wang also used the other three computations to assess the international container ports in his approach. Through four varied computations, the average ranking order scores of 12 international container ports were, respectively, A1: 1.25, A2: 7.75, A3: 4.25, A4: 5, A5: 3, A6: 1.75, A7: 5.75, A8: 9.25, A9: 12, A10: 9.75, A11: 7.25, and A12: 11. Based on the above, the average ranking order scores computed by Wang were different from the ranking order of Table 13. It was due to equipment and vessel shortages, total capacity decline, port congestion, and cost increase during COVID-19 outbreaks. Undoubtedly, COVID-19 outbreaks reserved the ranking orders of these international container ports from 2016 to the present. For instance, A1 in Wang's approach was the optimal port of the 12 international container ports, based upon on performance evaluation, whereas the ranking order for A1 was merely 6 in this paper. In fact, A1 indicated that the port was in Hong Kong. Because of COVID-19 outbreaks, the Hong Kong economy has deteriorated more than others due to numerous factors, and thus its port performed less well than before, too. The sort variations of other international container ports on performance evaluation could be discussed through the similar analysis. Evidently, the dilemma of epidemic prevention and economic development is an important issue for all governments. For a complicated environment, TIVFNs present more data than other fuzzy numbers. Therefore, international container ports with DEC, using the hybrid method of combining QFD with TOPSIS under IVFE, are effectively and efficiently assessed for service performance, and more messages are gained than in the past, meaning we can establish the essential fundamentals of port logistics related to COVD-19 barriers, and the recent scientific developments in port logistics due to the pandemic.

## 5. Conclusions

Since the evaluation of service performance of international container ports is a FMCDM problem with DEC, we extend QFD and TOPSIS under a fuzzy environment into IVFMCDM for the evaluating of the problem in this paper. Through EFPR, the multipli-

cation of IVFNs to form PFNs is not necessary for computations of extending QFD and TOPSIS into a hybrid method under a fuzzy environment. The hybrid method can deal with the FMCDM problems with DEC and avoid the corresponding drawbacks of ANP or IDEC transformed into DEC. Moreover, a port's relative closeness coefficients based on the EFPR are derived as the sorting reference for service performance evaluation of international container ports with DEC. Further, IVFMCDM provides the preference degrees of ports over the anti-ideal solution, and the ideal solution over ports on varied criterion besides relative closeness coefficients, so that managers, through the data in Table 7, Table 9, and Table 10, can evaluate ports based on different perspectives. Evidently, the IVFMCDM method, extending QFD and TOPSIS under a fuzzy environment, has three advantages: processing of decision-making problems with DEC, easy computation, and more message grabbing ability than other fuzzy numbers. Therefore, the service performance of international container ports with DEC are effectively and efficiently evaluated. In the future, varied data specifications may be in FMCDM problems with DEC because more messages are taken into consideration to solve decision-making problems. Decision-makers should match each kind of data specification to develop decision-making methods. The hybrid method of combining QFD with TOPSIS provides a logical underlying to propose a new perspective for management applications including scientific developments of port logistics.

**Author Contributions:** Conceptualization, Y.-J.W. and L.-J.L.; methodology, Y.-J.W.; validation, Y.-J.W., L.-J.L. and T.-C.H.; formal analysis, Y.-J.W. and T.-C.H.; investigation, Y.-J.W.; resources, T.-C.H.; data curation, Y.-J.W.; writing—original draft preparation, Y.-J.W.; writing—review and editing, Y.-J.W., L.-J.L. and T.-C.H.; visualization, L.-J.L.; supervision, Y.-J.W.; project administration, L.-J.L.; funding acquisition, Y.-J.W. All authors have read and agreed to the published version of the manuscript.

**Funding:** This research was partially funded by the Ministry of Science and Technology of the Republic of China under Grant No. MOST 108-2410-H-346-003.

**Institutional Review Board Statement:** Not applicable.

**Informed Consent Statement:** Not applicable.

**Data Availability Statement:** Not applicable.

**Acknowledgments:** Thanks for the partial support of the Ministry of Science and Technology of the Republic of China under Grant No. MOST 108-2410-H-346-003.

**Conflicts of Interest:** No conflict of interest exists.

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
