# Peer review of "Interval-Valued Fuzzy Multi-Criteria Decision-Making with Dependent Evaluation Criteria for Evaluating Service Performance of International Container Ports"

_jmse, doi:10.3390/jmse10070991_

Round 1

Reviewer 1 Report

Dear authors,

Thank you for your submission. I think your paper is an informative paper that can be published after a major revision.

Author Response

We are grateful for the constructive and helpful comments of reviewers. We had substantially revised the paper along these suggestions. The descriptions below provided a summary of how we had addressed the related comments.

Reviewer 2 Report

The paper “Interval-valued fuzzy multi-criteria decision-making with dependent evaluation criteria for evaluating service performance of international container ports” develops an evaluation method with effective and efficient in decision-making for assessing service performance of international container ports based on dependent evaluation criteria. As for the theoretical background, my suggestion is to clarify how the literature on the topic was reviewed to justify the research framework according to relevant theories in the body of literature. In fact, the authors should improve the theoretical justification of the study. Which is/are the theory/ies adopted to justify the study. I suggest to improve theoretical justification according to a relevant theory in the field of green management and guarantee that the authors have not overlooked some relevant papers in the analysis of the theoretical background. Moreover, the authors present an introduction section, in which the integration of social, economic and environmental issues should better analysed. In fact, in this present version this section neglects some valuable recent contributions in the body of knowledge, and therefore the authors should work more to update the literature background. It may be interesting to highlight in that regard that while service performance costs are immediate, their returns may be on the longer term (different scope in terms of time) and refer to issues considered as externalities for the single organisation (different scope in terms of actors involved in intra-organisational management process). As for the future contributions and implications, I suggest to highlight the potential role of blockchain as a future research direction for international container ports (see the contributions by Centobelli et al., Lim et al., Oropallo et al.; Liu et al.). I suggest to use also them as perspective of analysis in discussing the future research directions. I suggest to improve the discussion of findings. With regard to the context of investigation, could it affect the results? Why? Additional information on how the evaluation is generated should be included to justify the significance of the results and clarify the effectiveness and rationality of the proposed approach. Moreover, the author/s should describe more in details if and how their application converges or diverges from other approaches? The authors should improve the theoretical justification according to previous methodological contributions. Furthermore, the authors should enhance the practical implications for managers exploring the effects of extrinsic cue in the formation of quality perceptions. Finally, proofreading in different sections is needed.

Author Response

We are grateful for the constructive and helpful comments of reviewers. We had substantially revised the paper along these suggestions. The descriptions in the attachment provided a summary of how we had addressed the related comments.

Reviewer 3 Report

The manuscript "Interval-valued fuzzy multi-criteria decision-making with dependent evaluation criteria for evaluating service performance of international container ports" presents a timely topic in research, business and logistics, and the environment, especially the ports.

The scientific soundness of the current research is highly recommended. The authors did a very good job and analysis. However, the general quality of the presentation is scientifically embarrassing. 

If the following would be improved, the editors could accept the paper.

1. Thye aim and objectives of the study are missing or not clearly stated. This makes the paper lacking as it seems to offer no solution given the aim is not clearly stated.

2. Such a powerful article deserves a powerful discussion that is scientifically missing. The discussion should also be supported with various references and citations. Your current references and citations are scientifically insufficient. 

Author Response

(The authors gave the same response as above.)

Reviewer 4 Report

This article is relevant because the quality of seaport service is of great importance for optimizing logistics processes. The article title adequately reflects the content. In the abstract, the authors briefly give the article essence, describe the problem state, research methods, results. Key words correspond to the article content.

In the introduction, the authors describe the problems that have arisen in port logistics against the backdrop of COVID-19, the criteria that serve to evaluate the performance of ports, and also provides the article structure. The second section is devoted to the mathematical justification of the proposed method. In the third section, the authors describe a generalized method for multicriteria analysis. The fourth section contains a numerical example of the method proposed by the authors. In the "Conclusions" section, the authors summarize the obtained results.

The article is prepared in accordance with the instructions for the authors, corresponds to the topic that it explores and publishes. Theoretical and practical conclusions are supported by figures and tables that are of sufficient quality. The list of literary sources is sufficient, but it is desirable to supplement it with studies carried out in recent years, since it contains few references to modern sources.

In our opinion, the article is in line with the theme of "improving the sustainability and efficiency of maritime logistics".

Comment.

1. From the article text it is not clear how the criteria for evaluating the activities of seaports were selected and how the adequacy of the choice was checked.

2. In our opinion, it is necessary to more clearly formulate the purpose and objectives of the study, as well as to confirm how the methods used make it possible to assess the adequacy of the conclusions given in the article.

3. The authors argue that the proposed method will improve the efficiency of assessing the activities of seaports, but it is not clear on what basis such conclusions are obtained, and most importantly, how the proposed method will improve the efficiency of operational management in maritime logistics.

Author Response

We are grateful for the constructive and helpful comments of reviewers. We had substantially revised the paper along these suggestions. The descriptions in the attachment provided a summary of how we had addressed the related comments

Round 2

Reviewer 3 Report

The authors improved the paper but did not pay attention to my comment number 2 which makes this paper scientifically sound. Please see comment number two and add a section discussing of results, discuss the results and compare and contrast with previous studies.  Without the discussion of results are irrelevant.

Author Response

(The authors gave the same response as above.)

Round 3

Reviewer 3 Report

The authors have really improved the work and it is scientifically sound. For the importance of this paper on the subject, I would like to encourage you to support your discussion with more previous studies citing more of them as well as improving the scientific soundness.

Author Response

Dear Editors,

  Enclosed is the revised paper entitled “Interval-valued fuzzy multi-criteria decision-making with de-pendent evaluation criteria for evaluating service performance of international container ports” submitted to for this special issue entitled “Recent Scientific Developments in Port Logistics” of The Journal of Marine Science and Engineering - JMSE (ISSN 2077-1312) for possible publication. Thank you for processing. We are looking forward to the result of processing.

Sincerely Yours,

Yu-Jie Wang 1, Li-Jen Liu 2 and Tzeu Chen Han 1

1       Department of Shipping and Transportation Management, National Penghu University of Science and Technology, Penghu 880, Taiwan, ROC; [email protected], [email protected]

2       Dinos International Corporation, Taipei 104, Taiwan, ROC; [email protected]

Interval-valued fuzzy multi-criteria decision-making with de-pendent evaluation criteria for evaluating service performance of international container ports

jmse-1765829

Revisions in response to Editor’s comment

Comment

Respond

Please provide a point-by-point response to the reviewer’s comments and either enter it in the box below or upload it as a Word/PDF file. Please write down "Please see the attachment." in the box if you only upload an attachment. An example can be found here.

We are grateful for the constructive and helpful comments of the reviewer. We had substantially revised the paper along these suggestions. The descriptions below provided a summary of how we had addressed the related comments.

Revisions in respond to the comment of Reviewer #3

Comment 2

Respond

Such a powerful article deserves a powerful discussion that is scientifically missing. The discussion should also be supported with various references and citations. Your current references and citations are scientifically insufficient.

  The authors improved the paper but did not pay attention to my comment number 2 which makes this paper scientifically sound. Please see comment number two and add a section discussing of results, discuss the results and compare and contrast with previous studies. Without the discussion of results are irrelevant.

The discussion of comparing results with previous studies was described as follows.

“In 2016, Wang[39] evaluated these international container ports above by weak-ness and strength indices of FMCDM. Beside the weakness and strength indices of FMCDM, Wang also used the other three computations to assess these international container ports in his approach. Through varied four computations, average ranking order scores of twelve international container ports were respectively A1: 1.25, A2: 7.75, A3: 4.25, A4: 5, A5: 3, A6: 1.75, A7: 5.75, A8: 9.25, A9: 12, A10: 9.75, A11: 7.25, and A12: 11. Based on above, the average ranking order scores computed by Wang were different from the ranking order of Table 13. It was owing to equipment and vessel shortages, total capacity decline, port congestion, and cost in-crease during COVID-19 outbreaks. Undoubtedly, COVID-19 outbreaks reserved the ranking orders of these international container ports from 2016 to the recent. For instance, A1 in Wang’s approach was the optimal one for the twelve international container ports on performance evaluation, whereas the ranking order of A1 was merely 6 in this paper. In fact, A1 indicated that the port was in Hong Kong. For COVID-19 outbreaks, the economics of Hong Kong more deteriorated than others due to numerous factors, and thus its port performed less well than before, too. The sort variations of other international container ports on performance evaluation could be discussed through the similar analysis. Evidently, the dilemma of epidemic prevention and economic development is an important issue for all governments. To complicated environment, TIVFNs present more data than other fuzzy numbers. Therefore, international container ports with DEC by the hybrid method of combining QFD with TOPSIS under IVFE is effectively and efficiently assessed for service performance, and more messages are gained than the past to establish essential fundamentals in recent scientific developments of port logistics on account of defensing COVID-19 barriers.” are expressed on Pages 25 and 26.

I apologized for lacking descriptions in the previous manuscript. Thanks for your comment.
